

# Land cover effects on hydrologic services under a precipitation gradient

Ane Zabaleta[1], Eneko Garmendia[2, 3], Petr Mariel[4], Ibon Tamayo[5], Iñaki Antigüedad[1]

[1]Hidro-Environmental Processes Group, Science and Technology Faculty, University of the Basque Country UPV/EHU, Leioa, 48940, Basque Country, Spain
[2]Department of Applied Economics I, University of the Basque Country UPV/EHU, Vitoria-Gasteiz, 01006, Spain
[3]Basque Centre for Climate Change (BC3), Parque Científico de UPV/EHU, Leioa, 48940, Spain
[4]Department of Applied Economics III (Econometrics and Statistics), University of the Basque Country UPV/EHU, Bilbao, 48015, Spain
[2]Department of Statistics, Harvard University, Cambridge, 02140, US

*Correspondence to*: Ane Zabaleta (ane.zabaleta@ehu.eus)

**Abstract.** Climate change impacts on the hydrological cycle are altering the quantity, quality, and temporal distribution of riverine discharge, necessitating a more rigorous consideration of changes in land cover and land use. This study establishes relationships between different land cover combinations (e.g., percentages of forest – both native and exotic – and pastureland) and hydrological services, using hydrological indices estimated at annual and seasonal time scales in an area with a steep precipitation gradient (900–2600 mm y$^{-1}$). Using discharge data from 20 catchments in the Bay of Biscay, a climate transition zone, the study applied multiple regression models to better understand how the interaction between precipitation and land cover combinations influence hydrological services. Findings showed the relationship between land cover combinations and hydrological services is highly dependent on the amount of precipitation, even in a climatically homogeneous and relatively small area. In general, in the Bay of Biscay area, the greater presence of any type of forests is associated with lower annual water resources, especially with greater percentages of exotic plantations and high annual precipitation. Where precipitation is low, forests show more potential to reduce annual and winter high flows than pasturelands, but this potential decreases as annual or seasonal precipitation increases. As annual precipitation increases, low flows increase as the percentage of exotic plantations decreases and pasturelands increase. Results obtained in this study improve understanding of the multiple effects of land cover on hydrological services, and illustrate the relevance of land planning to the management of water resources, especially under a climate change scenario.

## 1 Introduction

The potential impacts of land cover on the hydrological cycle should be considered during land use policy-making (Fidelis and Roebeling, 2014; Ellison et al., 2017), including by integrating mitigation and adaptation strategies (Locatelli et al., 2016). Climate change impacts on the hydrological cycle are altering the quantity, quality, and temporal distribution of discharge in



rivers (Bates et al., 2008); however, climate change alone is insufficient to explain observed trends in streamflow (Schilling and Libra, 2003; Gallart and Llorens, 2003; Tomer and Schilling, 2009; López-Moreno et al., 2011). To fully understand these trends, changes in land cover and land use must be also considered (Garmendia et al., 2012, Liu et al., 2013; Brogna et al., 2017). Bauman et al. (2007) noted that vegetation is often the main driving force in ecosystem effects that influence

hydrological service provision. For example, in areas such as the Pyrenean region, observed decreases in streamflow have been related primarily to changes in land cover, rather than to climate change (Gallart and Llorens, 2004; López-Moreno et al., 2011; Morán-Tejeda et al., 2012).

Worldwide, deforestation rates outstrip afforestation by several million hectares per year. Overall global forest cover declined by 3.25% between 1990 and 2015 (FAO, 2016). During this same period, plantations increased globally by over 105 million

hectares; by 2015, about 31 % of the world's forests were designated primarily as production forests. This expansion is actively supported by governments (Enters and Durst, 2004; Schirmer and Bull, 2014), which assume that plantation forests can provide a range of economic, social, and environmental benefits. However, the impact of afforestation on water-related ecological services is not usually considered (Ellison et al., 2017).

Research suggests that forests play important roles in regulating fresh water flows (van Dijk and Keenan, 2007). Trees may

enhance soil infiltration and, under suitable conditions, improve groundwater recharge, delivering purified ground and surface water (Calder, 2005; Neary et al., 2009). Yet, the interpretation of the relationship between forests and hydrological services remains controversial (Bosch and Hewlett, 1982; Calder et al., 1997; Iroumé and Huber, 2002; Brown et al., 2005; Little et al., 2009; Keestra et al., 2018). Results vary among geographical latitudes and can be influenced by factors such as forest characteristics, changes in the seasonal structure, bio-geographical characteristics of catchments, soil types, and spatial scales.

Typically, however, streamflow decreases substantially following afforestation and reforestation, and increases after deforestation or forest clearing (Bosch and Hewlett, 1982; Andréassian, 2004; Farley et al., 2005, Li et al., 2017).

Streamflow changes are influenced by characteristics such as forest tree species, age, and density. A change from coniferous to deciduous forest cover can improve catchment water yield (Hirsch et al., 2011), presumably due to the longer rotation times. Young, fast-growing forests typically consume more water than old-growth forests (Kuczera, 1987; Vertessy et al., 2001;

Delzon and Loustau, 2005). Hence, though the impact of tree cover on water provision service tends to be negative in the short term, it may become neutral over the long term (Scott and Prinsloo, 2008). In the appropriate spatial settings, afforestation can improve water availability; however, plantation forests and the use of exotic species can disturb the hydrological balance with possible negative impacts (Trabucco et al., 2008 Huber et al., 2008; Lara et al., 2009, Little et al, 2009). Additionally, recent studies show that land management practices on tree plantations may promote rather than prevent soil erosion (Banfield et al.,

2018). This effect has been observed in the Bay of Biscay region (Zabaleta et al., 2016).

In the context of rising temperatures associated with climate change, afforestation may lead to additional decreases in available water resources (Rind et al., 1990; Liu et al., 2016). Thus, climate change mitigation policies focused on carbon sequestration could negatively affect water provision service (Jackson et al., 2005). These observations highlight the importance of placing water-related ecosystem services at the centre of reforestation and forest-based mitigation strategies, while considering carbon



storage and timber/non-timber forest products as co-benefits of strategies designed to protect the hydrological cycle (Locatelli et al. 2015; Ellison et al., 2017).

Efforts to prioritise hydrological services in mitigation and adaptation strategies, and, hence, in land use policy, should be supported by a solid understanding of how trees, forest characteristics, and forest management strategies influence water flows

in areas with different climatic, geographical, geological, and biological characteristics. Toward this end, this study analysed the effect of alternative land cover types (i.e., pastureland, native forests, and exotic plantations) on hydrological services in the Bay of Biscay using hydrological indicators obtained from the discharge series of 20 catchments during the periods 2000–2004 and 2007–2011. The specific objectives of this research were: i) to analyse the relationship between precipitation and several hydrological indicators at annual and seasonal time scales in an area with a steep precipitation gradient (Bay of Biscay);

(ii) to assess the relationships between different alternative land covers in use and hydrological indicators considering the existing precipitation gradient; and (iii) to detect patterns in the study area that should be considered in devising adaptation strategies and land use management policies.

## 2 Study area

The study area is in Gipuzkoa Province (1980 km$^2$) in the Basque Country of southwestern Europe (average latitude 43 °N

and average longitude 1 °W; Fig. 1). The latitude of the province and its geographical situation near the Bay of Biscay favour high mean annual precipitation (1500 mm with no dry season) and a mild climate (mean annual temperature of 13 °C) that varies little between winter (8–10 °C, on average) and summer (18–20 °C, on average). A spatial gradient is observed in annual precipitation, with maximum values in the eastern part of the province (up to 2500 mm) and decreasing precipitation towards the west and south (to about 1000 mm). The altitude ranges from sea level to a maximum of 1554 m; steep slopes, exceed 25

20  % throughout most of the province with mean values between 40 % and 50 % for most of the catchments.

The drainage network in the area is dense and can be described as dendritic and rectangular. The main rivers flow generally south to north, perpendicular to the coast line. Tributaries are frequently perpendicular to main rivers, influenced by the geological structure of the region, resulting in very narrow water courses that are typically short and steep. Gipuzkoa is located at the western end of the Pyrenees; the region is structurally complex and lithologically diverse, with materials from Palaeozoic

plutonic rocks to Quaternary sediments (EVE, 1990). Most of the materials in this region (>70 %) are of low or very low permeability. Sandstones, shales, limestones, and marls are dominant in most of the region, except in the east, where slates prevail (Zabaleta et al., 2016).

The mean soil depth in the study area is about 1 m, but highly variable. Cambisol is the prevailing soil type (FAO, 1977), generally characterised by a loam texture. Forests are the main land use (73 % in 2011) (MAGRAMA, 2013). The original

broad-leaved forests (oak–*Quercus robur*, and beech–*Fagus sylvatica*), presently reduced to 15 % of their original area, share space with tree plantations of rapid-growth exotic species such as *Pinus radiata*. These exotic species were introduced in the





second half of the twentieth century as a result of government support for afforestation policies and currently cover 39–48 % of the area that could sustain oak forests (Garmendia et al., 2012).

The abandonment of traditional cattle and sheep farming practices has also contributed to the conversion of pastureland and rangelands to fast-growth exotic plantations (Ruiz Urrestarazu, 1999). *Pinus radiata* stands in Gipuzkoa are well adapted to the environment and provide good support for the rapid development of forest communities (Carrascal, 1986; Ainz, 2008). Nevertheless, the expansion of these plantations results in substantial changes not only in the landscape, but also in forest management that affects the hydrological cycle (Garmendia et al., 2012) and sediment delivery (Zabaleta et al., 2016).

The study catchments exhibit a diverse mix of land cover types within a small geographical area that has similar climatic, geological, and topographical characteristics. This provides a good empirical basis for analysing how different land cover types affect median, low, and high flows. More precisely, the 20 selected catchments (some of which are nested catchments; see Fig. 1 and Table 1) include pasturelands (herbaceous vegetation), native forests (oak and beech), and exotic plantations (*Pinus radiata* plantations), as well as small areas of other cover types (e.g., urban land, roads, bare rock, water bodies, etc.). Discharge and precipitation are routinely measured for all the catchments as they are part of the hydro-meteorological monitoring network of Gipuzkoa Provincial Council (http://www.gipuzkoahidraulikoak.eus/es/).

## 3 Methodology

The methodology employed to assess the impacts of alternative combinations of different land cover types on selected hydrological indicators can be summarised in four steps: 1) extraction of hydrological indicators from discharge data series; 2) measurement of alternative land covers for each catchment; 3) assessment of the extent to which annual and seasonal precipitation control the hydrological indicators; and 4) analysis of the relationship between hydrological indicators, precipitation, and alternative land covers.

### 3.1 Hydrological data

Gauging stations included in the hydro-meteorological network of the Basque Country are located at each outlet of the 20 studied catchments (Fig. 1, Table 1). Water depth (m) is measured every 10 minutes and discharge ($m^3 s^{-1}$) is estimated through calibration conducted by the water services of the province (Environment and Hydraulic Works Department of Gipuzkoa Provincial Council) (Zabaleta et al., 2016).

To maintain coherence with land cover data obtained from forest inventories carried out during 2002 and 2009, discharge data was considered for two five-hydrological-year periods. The first period, from 2000–2001 to 2004–2005, was compared with land cover data obtained during 2002 (IFN3, 2005). The second period, from 2007–2008 to 2011–2012, was compared with land cover data from 2009 (IFN4, 2011). In this way, two sets of discharge series, accounting for a total of 10 hydrological-years, were selected for each gauging station. To facilitate comparison among catchment responses, all discharge data, including those for hydrological indicators, are referred to as specific discharges ($L s^{-1} km^{-2}$).





A comparison of the daily hydrographs obtained for the outlets of the 20 catchments shows the homogeneity in the timing of the discharge (Appendix A) and its relationship to the prevalence of Atlantic storms approaching from the northwest (Nadal-Romero et al., 2015). These storms influence the entire study area and are the main sources of precipitation. For this reason, even if there are important differences in total amounts of precipitation from east (higher) to west (lower), the distribution of precipitation over time is very similar in all the catchments analysed, which translates to similar patterns in the annual hydrographs.

Hydrological indicators related to different hydrological services were calculated considering fundamental characteristics of streamflow: magnitude, frequency, variability, and timing (Ritcher et al., 1996; Olden and Poff, 2003). As a first step, seven hydrological indicators were calculated from the discharge series for each hydrological year. At annual and seasonal time scales, the 10th (10m), 50th (50m), and 90th (90m) percentiles (L s$^{-1}$ km$^{-2}$) were analysed as indicators of discharge magnitude (Fig. A1). The coefficient of variation (CV) was used as a measure of the variability of the discharge series analysed. At the annual scale, the following were also calculated: runoff (R, mm), timing of low flows expressed as the first Julian day of the low flow period (J10), and skewness (skn), as a measure of the asymmetry of the hydrograph related to the frequency of discharge data, of each of the series. As a result, the annual value (Y) and the values for autumn (A), winter (W), spring (Sp) and summer (Su) were obtained for each hydrological year for the different indicators. All calculated indicators are listed in Table 2.

### 3.2 Land cover data

In 2005 and 2011, the Basque Government published detailed forest inventories. These forestry maps of the Basque Country were created by on-screen photo-interpretation, based on colour orthophotos generated by the SIGPAC project (http://www.mapama.gob.es/es/agricultura/temas/sistema-de-informacion-geografica-de-parcelas-agricolas-sigpac-/), with a minimum pixel size of 25 cm. For this study, geographic information systems were used to reclassify the land cover types into four main types: native forest, exotic plantations, pasturelands, and others. The areas corresponding to each type in each of the catchments were estimated using Environmental Systems Research Institute (ESRI) software (ArcGIS 10.1). The resulting data are listed in Table 1, which shows the percentage of each land cover type in the 20 catchments for both 5-year periods. Note that variations in land cover between the two periods are small.

### 3.3 Precipitation data

Annual precipitation (YP, mm) estimates for each of the 20 catchments were provided by the Environment and Hydraulic Works Department of the Gipuzkoa Provincial Council. These estimates were calculated by interpolation, based on a universal isotropic kriging method, using data obtained from the rain gauge network. Seasonal precipitation amounts for autumn (AP, mm), winter (WP, mm), spring (SpP, mm) and summer (SuP, mm) were also computed. These values were used to describe the overall precipitation regime for the 10 hydrological years under study and to assess the extent to which precipitation controlled the hydrological variables considered.



The annual and seasonal distribution of precipitation across catchments over the period studied is shown in Appendix B. In the catchments studied, annual precipitation varied from minimums of 958 mm for the 2001/2002 hydrological year in the C1Z2 catchment and 1581 mm for 2008/2009 in the C1P3 catchment to maximums that range from 1664 mm in D2W1 for 2001/2002 to 2611 mm in F1W1 for 2008/2009 (Fig. B1). On a seasonal basis (Fig. B2), autumn is usually the rainiest season

with a mean precipitation of 551 mm, followed by winter and spring with means of 425 mm and 343 mm respectively; summer is the season with the least precipitation (mean of 216 mm).

### 3.4 Statistical analysis

As a first step, the influence of precipitation on different hydrological indices was analysed using the following simple linear regression, Eq. (1):

$$H_{it} = \gamma_1 + (\gamma_2 \times P_{it}) + \varepsilon_{it} \tag{1}$$

where $H_{it}$ represents one of the hydrological indices in Table 2; $\gamma_1$ and $\gamma_2$ are parameters to be estimated; $P_{it}$ is the total precipitation amount considered; and $\varepsilon_{it}$ is a regression error satisfying the standard basic assumptions for each of the $i = 1,\ldots,$ 20 catchments and $t = 1,\ldots,$ 10 hydrological years.

Equation (1) was extended to include different land covers to study their possible influence on the hydrological indices, taking

into account their interactions with precipitation. Thus, Eq. (1) was extended to Eq. (2):

$$H_{it} = \beta_1 + (\beta_2 \times P_{it}) + (\beta_3 \times Nat_{it}) + (\beta_4 \times Exo_{it}) + (\beta_5 \times Past_{it}) + (\beta_6 \times Nat_{it} \times P_{it}) + (\beta_7 \times Exo_{it} \times P_{it}) +$$
$$(\beta_8 \times Past_{it} \times P_{it}) + \mu_{it} \tag{2}$$

where $\beta_1$ to $\beta_8$ are parameters to be estimated; $Nat_{it}$, $Exo_{it}$, and $Past_{it}$ are the percentages of native forest, exotic forest, and pastureland, respectively; and $\mu_{it}$ is a regression error satisfying the standard basic assumptions for each of the $i = 1,\ldots,$ 20

catchments and $t = 1,\ldots,$ 10 hydrological years.

Because Eq. (2) includes interactions of explanatory variables, the interpretation of results can be difficult. For this reason, different representative combinations were defined based on real values of explanatory variables and the corresponding predictions of hydrological indices were computed. This allowed for a simple and direct interpretation of the influence of all variables.

The objective of this study was to compare predicted hydrological indices for various land cover combinations under different precipitation amounts. To avoid considering outliers, the 1st and 3rd quartiles of the precipitation data series were calculated for the selected period (annual or seasonal) and defined as the low and high precipitation conditions. For annual scale data, annual precipitation was considered, while for seasonal scale, precipitation of the season studied plus that of the previous season (6 months total) were considered.

The different land cover combinations shown in Table 3 were explored under low and high precipitation conditions, and compared to a "base" land cover combination (combination 0) of 76 % exotic, 18 % pastureland, and 6 % native. Land cover combination 0 was defined as a combination with a maximum area of exotic plantations, minimum area of native forests, and a low percentage of pasturelands (calculated as the remaining percentage to cover 100 % of the area). Combinations from 1 to



5 were defined as realistic alternative patterns to combination 0. Differences between these patterns and combination 0 were calculated for each hydrological index under low and high precipitation conditions (Tables 4, 5 and 6). These combinations were defined considering real data (e.g., maximum, minimum, or mean percentages of native forests, exotic plantations, and pasturelands). Defined in this way, each combination was used to examine interactions between realistic data; results for scenarios that might be very different from the existing ones were not extrapolated.

## 4 Results and discussion

### 4.1 Effect of precipitation on hydrological indicators

Precipitation is generally agreed to be the main driver of large-scale variability in monthly, seasonal, and annual streamflows (Ward and Trimble, 2004). In the study area a certain spatial homogeneity in the precipitation-runoff ratio at an annual scale can be deduced from the high value of the coefficient of determination ($R^2 = 0.8$; $p$ value $< 0.001$) obtained in the linear regression (Eq. (1)) between annual precipitation (YP, mm) and annual runoff (YR, mm) (Fig. C1). The regression includes all data collected during the study period in the 20 catchments listed in Table 1, which constitutes 182 pairs of data (some pairs are not included in the analysis due to missing data) and has a high level of significance. Thus, precipitation explains a high percentage of the variability in annual runoff (80 %). There is also a significant correlation between annual precipitation and median, high, and low flows, with coefficients of determination of 0.75, 0.62 and 0.54 ($p$ values $<0.001$), respectively. Conversely, the relationships between precipitation and hydrological indicators related to variability (CV, at all time-scales), timing (JY of the beginning of low flows, JY10m), and frequency (skn) show very low coefficients of determination ($<0.25$) at an annual scale, and less than 0.1 at the seasonal scale in the case of the coefficient of variation. The significance of the relationship between seasonal precipitation and seasonal median, high, and low flows is lower than that at the annual scale: the coefficient of determination is greater than 0.5 ($p$ values $<0.001$) for median flows in spring and summer (Sp50m and Su50m; $R^2 = 0.58$ and 0.56, respectively), high flows in autumn and winter (A90m and W90m; $R^2 = 0.50$ and 0.55, respectively), and low flows in spring and summer (Sp10m and Su10m; $R^2 = 0.5$ and 0.51, respectively) (Figs. C2, C3, C4).

### 4.2 Effect of land cover on median discharge

Figure 2a shows results of the multiple regression, defined in Eq. (2), between alternative land covers, annual precipitation amounts, and median annual discharge as a hydrological index. The three land cover types are significant ($p$ values $< 0.05$) and precipitation is significant in interactions with all land covers ($p$ values $< 0.1$). This indicates that the degree of influence of precipitation is contingent on the specific land cover. The coefficient of determination is 0.78, indicating that the model fits the data well.

The results shown in Table 4 are expressed as the percentage change in the hydrological index with respect to the results obtained for the base land cover (combination 0) for low and high YP. The variations are highly conditioned by annual precipitation, both in terms of the percentage change and whether the change was positive or negative. Median annual discharge





(Y50m) increases when a decrease in exotic plantations is accompanied by an increase in pasturelands (combinations 1 and 4), by as much as 44 % and 67 % for low and high annual precipitation amounts, respectively. Conversely, replacing exotic plantations with native forests (combinations 2 and 3) has a slightly negative impact on median discharge for low precipitation amounts (up to 18 %) while for higher precipitation amounts median discharge increases (up to 24 %). Further, increases in

median discharge are higher with higher annual precipitation amounts. The magnitude of the observed change is similar to that reported by Farley et al. (2005) for catchments located in northern Europe and higher than those obtained from hydrological modelling by Carvalho-Santos et al. (2016) and Morán-Tejeda et al. (2014) for catchments in the Iberian Peninsula.

Hence in the study area, greater forest cover may result in lower annual water yields. Similarly, numerous studies have shown that replacement of forest by grasslands leads to increased annual water yields, while afforestation processes can decrease

annual yields (e.g., Bosch and Hewlett, 1982; Brown et al., 2005; Brogna et al, 2017). Additionally, as annual precipitation amounts increase, this effect becomes clearer in the case of exotic plantations (Fig. 2). Forest plantation species have been selected for rapid early growth, which has high associated water consumption (Farley et al., 2005); maximizing timber production generally involves harvesting trees before their growth slows, that is, before their water consumption starts to decrease. Current forest management of cultivated plantations in the study area involves clearcutting with rotations of around

30 years. Conversely, native forests, established for other purposes, may be left to mature and hence will tend to exhibit lower water consumption (van Dijk and Keenan, 2007) and have reduced interception losses during the leafless period (autumn-winter).

Alternative land covers seem to have little significant effect on median discharge during autumn, winter, and summer, as the coefficients for land covers obtained in the multiple regressions are not significant (not shown). Nevertheless, the inclusion of

land cover is important for median spring discharge (Table 6), with effects similar to those observed for median annual discharge (Y50m). Regression results shown in Fig. 3b indicate that native forests, pasturelands, and precipitation in interactions with both types of land cover are significant ($p$ values <0.05). The percentage of exotic plantations in the catchment significantly influences the median discharge in the spring. The coefficient of determination for this graph is 0.63, indicating a good fit.

Decreasing the percentage of exotic plantations (by increasing pasturelands or native forests) increases spring discharge under high precipitation amounts (Sp50m). The magnitude of this increase is larger when the extension of pasturelands is larger (combinations 1 and 4). At low precipitation rates, the positive effect of pasturelands on Sp50m remains, while increasing native forest (combinations 2 and 3) has a negative effect on median discharge.

The base land cover combination used in Table 3 (with the highest percentage of exotic plantations) is associated with the least

change in median discharge indices across the precipitation gradient (combination 0, Fig. 2): Y50m and Sp50m vary from about 2–3 L s$^{-1}$ km$^{-2}$ for annual and seasonal precipitation amounts around 1000 and 400 mm, respectively, to about 20 L s$^{-1}$ km$^{-2}$ for precipitations around 2000 and 1000 mm, respectively. Conversely, land cover combinations 3 and 4 (with the highest percentage of native forests and pasturelands, respectively, and lowest percentage of exotic plantations) show the highest variation in Y50m and Sp50m in the precipitation gradient existing in the study area. Consequently, the land cover combination





that gives the highest or lowest median discharge values changes with annual and seasonal precipitation amounts. This fact must be kept in mind in an area with a steep precipitation gradient under current and future climate change scenarios.

### 4.3 Effect of land cover on high flows

As shown in Table 4, an increase in high flows (Y90m) can be observed at low precipitation amounts, as the percentage of
exotic plantations decreases and native forests (combination 3) or pasturelands (mainly, combination 4) increases. The increase in Y90m seems to be similar to the increase in native forest or in pastureland. For higher precipitation amounts, Y90m changes little with land cover, with the observed changes being negative in all cases. Considering changes in Y90m across the precipitation gradient for different land cover combinations (Fig. 3a), the base land cover combination (the one with the highest percentage of exotic plantations) exhibits the lowest Y90m for low annual precipitation, but it is also the one with the steepest
slope; hence, it is the combination that yields the highest Y90m results for higher precipitation. Conversely, combination 4, with the lowest percentage of exotic plantations considered (10 %), a moderate percentage of native forest (31 %), and high percentage of pastureland (59 %) exhibits the highest Y90m for low precipitation amounts and the lowest Y90m for higher ones. This indicates that the potential of forests to reduce high flows decreases as annual precipitation increases (Fig 3a), and therefore, in the area studied, when high annual precipitation is considered, this potential is quite low. Robinson et al., 2003
found that, under realistic forest management procedures, the potential for forests to reduce peak flows in Europe was lower than usually claimed.

Similar conclusions can be reached from seasonal data analysis. Land cover coefficients are significant only for the winter period; during autumn, spring, and summer, land cover does not appear as a significant variable influencing the magnitude of high flows. For winter (W90m), results differ depending on precipitation (in this case considered as the sum of winter and
previous autumn precipitation). Under low precipitation amounts, W90m increases (up to 31 %) in combinations (e.g., cases 3 and 2) where the decrease in exotic plantations is compensated for mainly by an increase in native forests (Table 5). This implies that high flows are attenuated under land cover combinations with high percentages of exotic plantations, which is favourable for flood regulation (Carvalho-Santos et al., 2016). Under high precipitation amounts, the situation remains practically unchanged for all land cover combinations.

### 4.4 Effect of land cover on low flows

With regard to satisfying aquatic ecosystem or socio-economic water demands, and, in turn, the ecological status of water bodies (European Commission, 2000), it is important to consider low flow values. For annual low flows, a small variation in Y10m (Table 4) was observed when the percentage of pasturelands increases at the expense of exotic plantations (combination 1), and a decrease in Y10m (up to 50 %) in combination 3 with 66 % of native forests under the lowest annual precipitation
amount. In contrast, for higher YP values, the land cover combination with the higher percentage of exotic plantations (combination 0) is one of the combinations with the lowest Y10m. Under high precipitation amounts (Table 4), low flows exhibit least change when decreases in exotics are compensated for by increases in native forests (combinations 2, 3, 5) and





Y10m increases when exotic plantations decrease and pasturelands increase (combinations 1, 4). Therefore, in line with the findings of Brogna et al., (2017), this study shows exotic plantations have a slightly positive effect on low flows under low annual precipitation amounts; with annual precipitation of less than 700 mm, other land covers provide smaller values of Y10m than the base combination. These positive effects disappear, however, under higher annual precipitation regimes. The positive

effects of forests on base flow, strongly related to annual low flows, have been associated with better infiltration of forested soils (Price, 2011), while negative effects have been linked to higher evapotranspiration rates (Hicks et al., 1991).

During winter, low flows (W10m) increase as exotic forests decrease in all land cover combinations and as native forests (combination 3) or pasturelands (combination 4) increase; the increase for W10m is greater with higher precipitation amounts (Table 5). During spring, native forests (combinations 2, 3) seem to have a negative effect (up to 87 %) on Sp10m when

precipitation is low, but this negative effect disappears in areas with higher precipitation (Table 6). Greater pastureland (combinations 1 and 4) positively affects springtime low flows under low and high precipitation rates by as much as 90 %. Land cover combination 3, which has the highest percentage of native forests (66 %) shows the greatest change in Y10m across the precipitation gradient of the study area (Table 4, Fig. 3b), while catchments with high percentages of exotic plantations (combination 0), show the least change in low flows across the precipitation gradient.

No statistically significant influences were observed on median, high, or low autumn and summer flows or on other hydrological indices related to the timing of low flows or other changes to the hydrograph; however, this does not rule out the possibility of relationships between land cover and the hydrological indices. As shown in Appendix C, precipitation (volume and distribution) is the main driver of the system, and thus the influence of other drivers, such as land cover, may fail to emerge as statistically significant.

Further study is needed in the Bay of Biscay area to determine how the characteristics of specific tree species (e.g., their phenology and physiology) affect various components of the hydrological cycle. Analyses are also needed to establish the relationship between forest types, land management issues and soil development. For instance, clearcutting of exotic species in the study area is usually accompanied by harvesting with chainsaws, skidding, and mechanical site preparation (prior to replanting) such as scarification and ripping (Gartzia-Bengoetexea et al., 2009). These logging operations alter the physical

properties of soil, affecting processes such as infiltration, evapotranspiration, percolation, and lateral flow, and in turn, catchment water balance and temporal distribution river discharge. A deepened understanding of those relationships will help achieve a solid understanding of how trees characteristics, forest types and management strategies, and soil properties influence water flows.

## 5 Conclusions

This study identifies the relationships among different land cover combinations (forests–native and exotic–and pasturelands) and hydrological services in an area with a steep precipitation gradient (900–2600 mm yr$^{-1}$). Annual and seasonal hydrological indices were estimated using discharge data from 20 catchments in the Bay of Biscay area. Results indicate that precipitation



has a significant positive impact on median, high, and low flows and is the main driver of annual and seasonal discharge. That strong influence may obscure the relationship between land cover and the hydrological responses of catchments in high precipitation gradient areas. From a policy-making perspective, it is important to assess how land cover changes affect streamflows, as these changes are strongly influenced by human intervention (e.g., through land use planning or public policies

to enhance certain land uses and constrain damaging practices, etc.).

Unravelling the effects of land cover on hydrological services is especially important in a climatic transition zone like the Bay of Biscay (Meaurio et al., 2017), which is characterised by a steep precipitation gradient and is subject to the uncertain effects of climate change in terms of magnitude and temporal distribution of precipitation projections. In this regard, the methodology developed in this study to deal with the interactions between the two drivers (i.e., precipitation and land cover) increases the

understanding of how various land cover combinations affect hydrological services across an entire precipitation gradient.

Conclusions about the effect of each land cover combination cannot be drawn without considering the amount of precipitation. However, results show that in the Bay of Biscay area, the presence of any kind of forest decreases annual water resources (Y50m), and this effect is more evident with exotic plantations as the annual precipitation increases. Similar to studies by Robinson et al. (2003), this study indicates that the potential for forests to reduce peak flows is lower than usually claimed;

however, the effect of land cover on high flows also changes with precipitation. For low precipitation amounts, forests, especially exotic plantations, show greater potential to reduce annual and wintertime high flows than pasturelands, but this potential decreases as annual or seasonal precipitation increases. Moreover, when high annual precipitation is considered, the potential of exotic plantations to reduce high flows is lower than that of native forests or pasturelands. The results also show that exotic plantations have a slight positive effect on annual low flows under low annual precipitation conditions; however,

low flows increase as annual precipitation increases and when exotic forests are replaced by pasturelands. This effect is most evident in winter and spring, and when the combination of pasturelands and native forests account for most of the catchment area.

Results from in this study show that a trade-off among the different hydrological services may emerge as a result of changes in land cover, and that such services are highly dependent on the amount of precipitation. Hence, to design appropriate water

management policies (e.g., to ensure the provision of water resources or to avoid the impact of extreme events), policy-makers need to focus on catchment-scale measures that consider the effect of land cover on hydrological services across a precipitation gradient. There simply is no unique "best combination" for all locations and all services. This is especially relevant under a climate change scenario, as precipitation projections remain largely uncertain both in magnitude and direction (e.g., positive or negative changes). It is time for land planning and forest policies to place water at the centre of the decision-making agenda.

ACKNOWLEDGEMENTS

This research was supported by the Basque Government (Consolidated Groups IT1029/16, IT-642-13) and the Spanish Ministry of Economy and Competitiveness (ECO2017-82111-R). Eneko Garmendia acknowledges financial support from




SOSTEPASTOS project (MIMECO AGL2013-48361-C2-1-R). The authors would like to thank Iñaki Aizpuru for his help with land cover data and the Environment and Hydraulic Work Department of the Gipuzkoa Provincial Council for providing hydrologic data series.

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



**Tables**

**Table 1.** Catchment descriptions.

| | | Catchment area (km²) | Primary Cover Types (%) | | | | | | | |
| | | | 2002 | | | | 2009 | | | |
| | | | Forest Cover | | | Pasture Cover | Forest Cover | | | Pasture Cover |
| Code | Catchment | | Native | Exotic | **Total** | | Native | Exotic | **Total** | |
| A1Z1 | San Prudentzio | 121.78 | 21 | 49 | **70** | 24 | 21 | 46 | **67** | 26 |
| A1Z2 | Oñati | 105.78 | 27 | 46 | **73** | 25 | 27 | 45 | **72** | 26 |
| A1Z3 | Urkulu | 9 | 43 | 14 | **57** | 42 | 47 | 10 | **57** | 42 |
| A2Z1 | Aixola | 5.03 | 6 | 76 | **82** | 12 | 7 | 73 | **80** | 15 |
| A3Z1 | Altzola | 464.25 | 19 | 52 | **71** | 23 | 20 | 50 | **70** | 24 |
| B1T1 | Barrendiola | 3.8 | 48 | 18 | **66** | 0 | 48 | 18 | **66** | 0 |
| B1Z1 | Aitzu | 56.13 | 19 | 56 | **75** | 18 | 19 | 55 | **74** | 19 |
| B1Z2 | Ibai-Eder | 62.73 | 25 | 53 | **78** | 20 | 25 | 53 | **78** | 20 |
| B2Z1 | Aizarnazabal | 269.77 | 19 | 50 | **69** | 26 | 20 | 49 | **69** | 26 |
| C1P3 | Arriaran | 2.77 | 24 | 62 | **86** | 12 | 24 | 62 | **86** | 12 |
| C1Z2 | Estanda | 55.02 | 16 | 54 | **70** | 23 | 17 | 52 | **69** | 24 |
| C2Z1 | Agauntza | 69.64 | 57 | 24 | **81** | 18 | 57 | 23 | **80** | 18 |
| C5Z1 | Alegia | 333.34 | 30 | 39 | **69** | 27 | 30 | 36 | **66** | 29 |
| C7Z1 | Berastegi | 33.34 | 25 | 35 | **60** | 38 | 25 | 34 | **59** | 37 |
| C8Z1 | Leitzaran | 110.01 | 39 | 39 | **78** | 20 | 46 | 33 | **79** | 20 |
| C9Z1 | Lasarte | 796.5 | 32 | 34 | **66** | 30 | 33 | 31 | **64** | 31 |
| D1W1 | Añarbe | 47.69 | 66 | 19 | **85** | 15 | 66 | 19 | **85** | 15 |
| D2W1 | Ereñozu | 218.42 | 48 | 30 | **78** | 21 | 50 | 27 | **77** | 21 |
| E1W1 | Oiartzun | 56.6 | 26 | 33 | **59** | 35 | 31 | 28 | **59** | 35 |
| F1W1 | Endara | 6.19 | 15 | 53 | **68** | 32 | 15 | 53 | **68** | 32 |

Note: Only forest (native and exotic) or pasture land covers are shown. Other types of land cover are not shown as they are usually less than 10 %, except for Barrendiola catchment where about 30 % of the catchment is bare rock.





**Table 2.** Statistics of hydrological indices, land cover types, and precipitation amounts used in this study. See the text for abbreviation meaning.

|  | Index | Mean | s.d. | Min | Max |
|---|---|---|---|---|---|
| Hydrologic indices | YR, mm | 839 | 320 | 246 | 1873 |
|  | Y10m, L s$^{-1}$ km$^{-2}$ | 4.93 | 4.40 | 0.14 | 28.22 |
|  | A10m, L s$^{-1}$ km$^{-2}$ | 6.90 | 7.75 | 0.28 | 67.48 |
|  | W10m, L s$^{-1}$ km$^{-2}$ | 13.55 | 10.31 | 1.91 | 90.89 |
|  | Sp10m, L s$^{-1}$ km$^{-2}$ | 8.42 | 6.63 | 0.34 | 36.18 |
|  | Su10m, L s$^{-1}$ km$^{-2}$ | 4.41 | 4.42 | 0.02 | 36.22 |
|  | Y50m, L s$^{-1}$ km$^{-2}$ | 14.68 | 9.72 | 2.71 | 58.98 |
|  | A50m, L s$^{-1}$ km$^{-2}$ | 18.94 | 16.63 | 2.18 | 118.74 |
|  | W50m, L s$^{-1}$ km$^{-2}$ | 27.22 | 15.63 | 5.59 | 125.77 |
|  | Sp50m, L s$^{-1}$ km$^{-2}$ | 17.50 | 12.29 | 2.85 | 66.43 |
|  | Su50m, L s$^{-1}$ km$^{-2}$ | 6.22 | 6.26 | 0.21 | 49.69 |
|  | Y90m, L s$^{-1}$ km$^{-2}$ | 59.81 | 24.62 | 15.27 | 153.27 |
|  | A90m, L s$^{-1}$ km$^{-2}$ | 74.40 | 45.14 | 7.22 | 217.21 |
|  | W90m, L s$^{-1}$ km$^{-2}$ | 87.18 | 34.28 | 17.38 | 190.39 |
|  | Sp90m, L s$^{-1}$ km$^{-2}$ | 49.17 | 24.24 | 7.61 | 113.48 |
|  | Su90m, L s$^{-1}$ km$^{-2}$ | 12.54 | 16.09 | 0.8 | 120.83 |
|  | CVY | 1.54 | 0.42 | 0.63 | 3.02 |
|  | CVA | 1.40 | 0.59 | 0.17 | 3.08 |
|  | CVW | 1.11 | 0.41 | 0.15 | 2.44 |
|  | CVSp | 1.17 | 0.53 | 0.36 | 3.54 |
|  | CVSu | 0.79 | 0.60 | 0.1 | 3.95 |
|  | skn | 4.57 | 2.00 | 0.92 | 12.43 |
|  | JY10m | 251 | 46 | 52 | 343 |
| Land cover | Forest, % | 72 | 8 | 56 | 85 |
|  | Native, % | 31 | 15 | 6 | 66 |
|  | Exotic, % | 41 | 16 | 10 | 76 |
|  | Pasturelands, % | 23 | 9 | 0 | 42 |
|  | Others, % | 5 | 7 | 0 | 34 |
| Precipitation | YP, mm | 1538 | 345 | 958 | 2611 |
|  | AP + SuP, mm | 776 | 228 | 323 | 1681 |
|  | WP + AP, mm | 975 | 253 | 441 | 1874 |
|  | SpP + WP, mm | 767 | 171 | 402 | 1406 |
|  | SuP + SpP, mm | 560 | 179 | 285 | 1146 |





**Table 3.** Percentage of different land cover types for each land cover combination. Note that base land cover combination in the text refers to combination (0).

| Land use combination | 0 | 1 | 2 | 3 | 4 | 5 |
|---|---|---|---|---|---|---|
| Exotic (%) | 76 | 40.8 | 40.8 | 10 | 10 | 40.8 |
| Native (%) | 6 | 6 | 41.2 | 66 | 30.84 | 30.84 |
| Pasture (%) | 18 | 53.2 | 18 | 24 | 59.16 | 28.36 |





**Table 4.** Results obtained, from multiple regression models, in the variation in percentage with respect to the base land cover combination (0) in the annual median, high and low flows with land cover for low and high precipitation amounts.

| | | Result for base land cover ($L\,s^{-1}\,km^{-2}$) | Differences from base land cover | | | | | Result for base land cover ($L\,s^{-1}\,km^{-2}$) | Differences from base land cover | | | | |
|---|---|---|---|---|---|---|---|---|---|---|---|---|---|
| | | Low Annual Precipitation (1279 mm) | | | | | | High Annual Precipitation (1719 mm) | | | | | |
| | Combination | 0 | 1 | 2 | 3 | 4 | 5 | 0 | 1 | 2 | 3 | 4 | 5 |
| Hydrologic differences | Y50m | 7.81 | 44 % | −15 % | −18 % | 41 % | 2 % | 14.59 | 52 % | 9 % | 24 % | 67 % | 22 % |
| | Y90m | 39.1 | 23 % | 17 % | 33 % | 39 % | 19 % | 70.87 | −11 % | −2 % | −5 % | −10 % | −3 % |
| | Y10m | 2.93 | −4 % | −31 % | −53 % | −27 % | −22 % | 6.25 | 45 % | −9 % | −8 % | 46 % | 7 % |

Note: Only results for regression with statistically significant coefficients for land cover variables and determination coefficient higher than 0.5 are included. Precipitation considered (1st and the 3rd quartiles of annual and seasonal) in each case is also included.





**Table 5.** Results obtained, from multiple regression models, in the variation in percentage with respect to the base land cover combination (0) in the winter high and low flows with land cover for low and high precipitation amounts.

| | | Result for base land cover ($L\ s^{-1}\ km^{-2}$) | Differences from base land cover | | | | | Result for base land cover ($L\ s^{-1}\ km^{-2}$) | Differences from base land cover | | | | |
|---|---|---|---|---|---|---|---|---|---|---|---|---|---|
| | | Precipitation = 814 mm (Winter + previous Autumn) | | | | | | Precipitation = 1147 mm (Winter + previous Autumn) | | | | | |
| | Combination | 0 | 1 | 2 | 3 | 4 | 5 | 0 | 1 | 2 | 3 | 4 | 5 |
| Hydrologic differences | W90m | 67.7 | –5 % | 18 % | 31 % | 8 % | 12 % | 106.34 | –7 % | 2 % | 3 % | –6 % | –0.4 % |
| | W10m | 8.5 | 19 % | 21 % | 39 % | 37 % | 21 % | 10.94 | 42 % | 40 % | 75 % | 77 % | 41 % |

Note: Only results for regression with statistically significant coefficients for land cover variables and determination coefficient higher than 0.5 are included. Precipitation considered (1st and the 3rd quartiles of annual and seasonal) in each case is also included.





**Table 6.** Results obtained, from multiple regression models, in the variation in percentage with respect to the base land cover combination (0) in the spring median and low flows with land cover for low and high precipitation amounts.

| | | Result for base land cover ($L\ s^{-1}\ km^{-2}$) | Differences from base land cover | | | | | Result for base land cover ($L\ s^{-1}\ km^{-2}$) | Differences from base land cover | | | | |
|---|---|---|---|---|---|---|---|---|---|---|---|---|---|
| | | Precipitation = 646 mm (Spring + previous Winter) | | | | | | Precipitation = 852 mm (Spring + previous Winter) | | | | | |
| | Combination | 0 | 1 | 2 | 3 | 4 | 5 | 0 | 1 | 2 | 3 | 4 | 5 |
| Hydrologic differences | Sp50m | 9.95 | 55 % | −32 % | −45 % | 42 % | −6 % | 15.6 | 56 % | 15 % | 35 % | 76 % | 27 % |
| | Sp10m | 4.98 | 69 % | −58 % | −87 % | 40 % | −20 % | 8.12 | 77 % | 0 % | 13 % | 90 % | 22 % |

Note: Only results for regression with statistically significant coefficients for land cover variables and determination coefficient higher than 0.5 are included. Precipitation considered (1st

5    and the 3rd quartiles of annual and seasonal) in each case is also included.





**Figure Captions**

**Figure 1.** a) Location and digital terrain model of the study area with the main drainage network, location of gauging stations, catchments, and average precipitation values. b) Land cover map of the study area in 2002 and 2009.

**Figure 2.** Expected values of a) annual average flows (Y50m, L s$^{-1}$ km$^{-2}$) and b) average discharge for spring (Sp50m, L s$^{-1}$ km$^{-2}$) for the land cover combinations described in Table 3 and a gradient of precipitation, as a result of the multiple regression models shown in Appendix D (tables D1 and D2, respectively).

10  **Figure 3.** Expected values of a) annual high flows (Y90m, L s$^{-1}$ km$^{-2}$) and b) low flows for spring (Sp10m, L s$^{-1}$ km$^{-2}$) for the land cover combinations described in Table 3 and a gradient of precipitation, as a result of the multiple regression models shown in Appendix D (tables D3 and D4, respectively).



## Appendix

**Appendix A:** Hydrographs for the 20 catchments in Table 1 for the hydrological year 2000–2001. The meaning of some of the calculated hydrological indicators is also indicated in the figure.

**Appendix B:** Boxplots representing the statistics of 1) annual and 2) seasonal precipitation for the 20 studied catchments during the hydrological years considered. A = Autumn, W = Winter, Sp = Spring and Sm = Summer.

**Appendix C:** Linear regressions obtained between 1) annual precipitation (YP, mm) and runoff (YR, mm) 2) precipitation
10 from spring and winter (SpP + WP) and average discharge in spring (Sp50m) 3) precipitation from winter and autumn (WP + AP) and wintertime high flows (W90m) and d4) precipitation from spring and winter (SpP + WP) and low flows in spring (Sp10m).

**Appendix D:** Multiple regression models. 1) Multiple regression model for annual average flows (Y50m) considering
15 alternative land cover and its interaction with annual precipitation (YP). 2) Multiple regression model for average discharge for spring (Sp50m) considering alternative land cover and its interaction with seasonal precipitation (spring + winter precipitation, SpPt). 3) Multiple regression model for annual high flows (Y90m) considering alternative land cover and its interaction with annual precipitation (YP). 4) Multiple regression model for low flows for spring (Sp10m) considering alternative land cover and its interaction with seasonal precipitation (spring + winter precipitation, SpPt).



**Figure 1.**





**Figure 2.**

a)

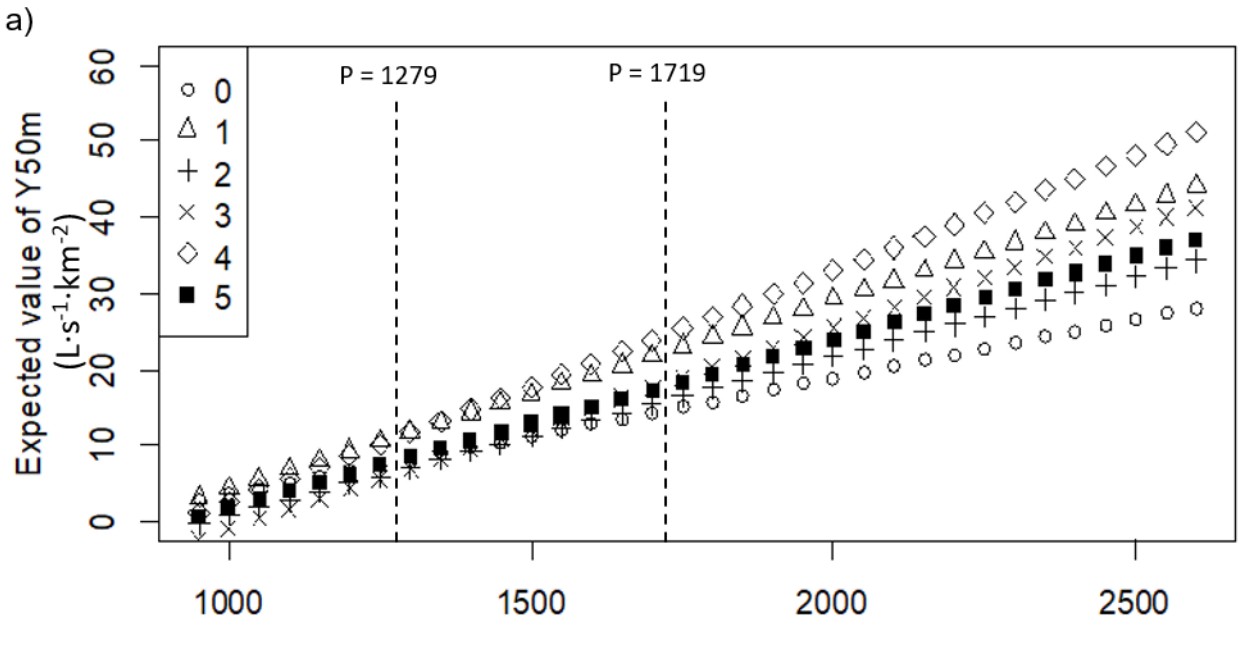

b)

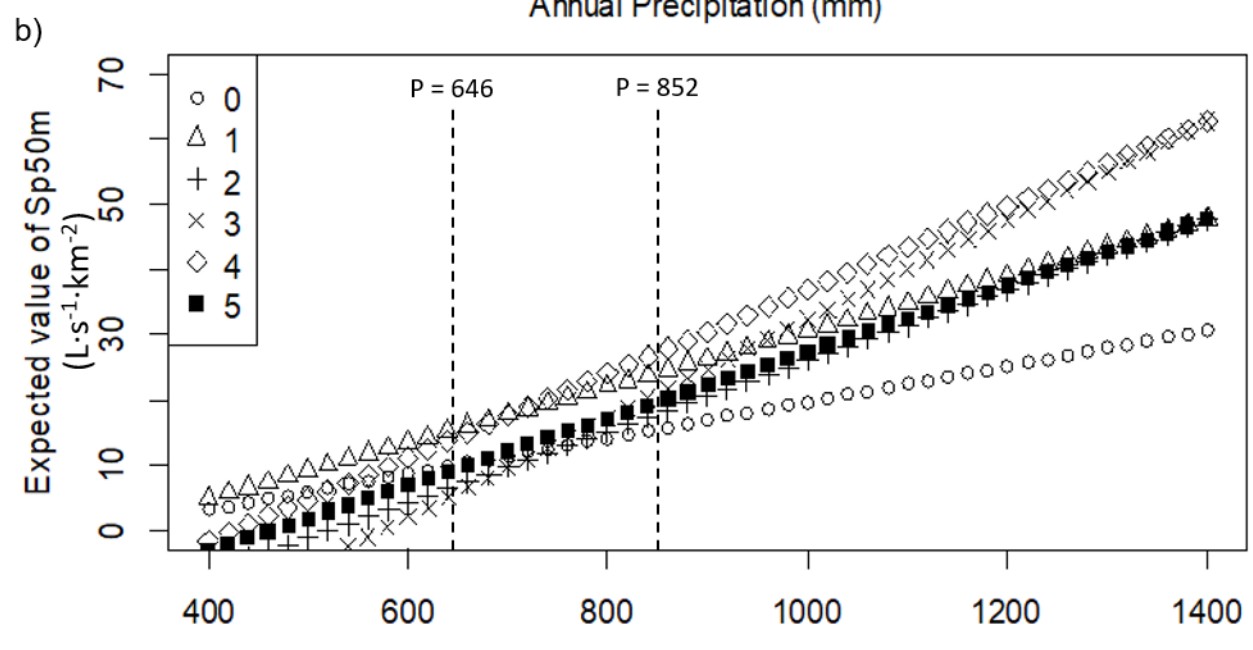



**Figure 3.**

a)

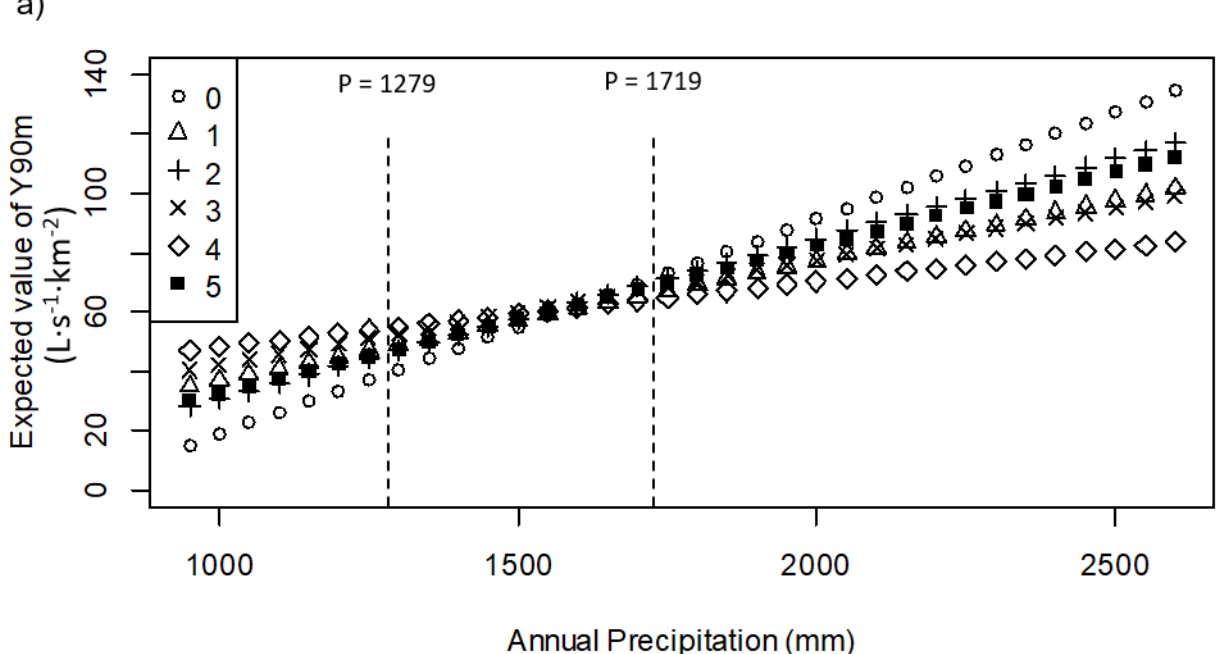

b)

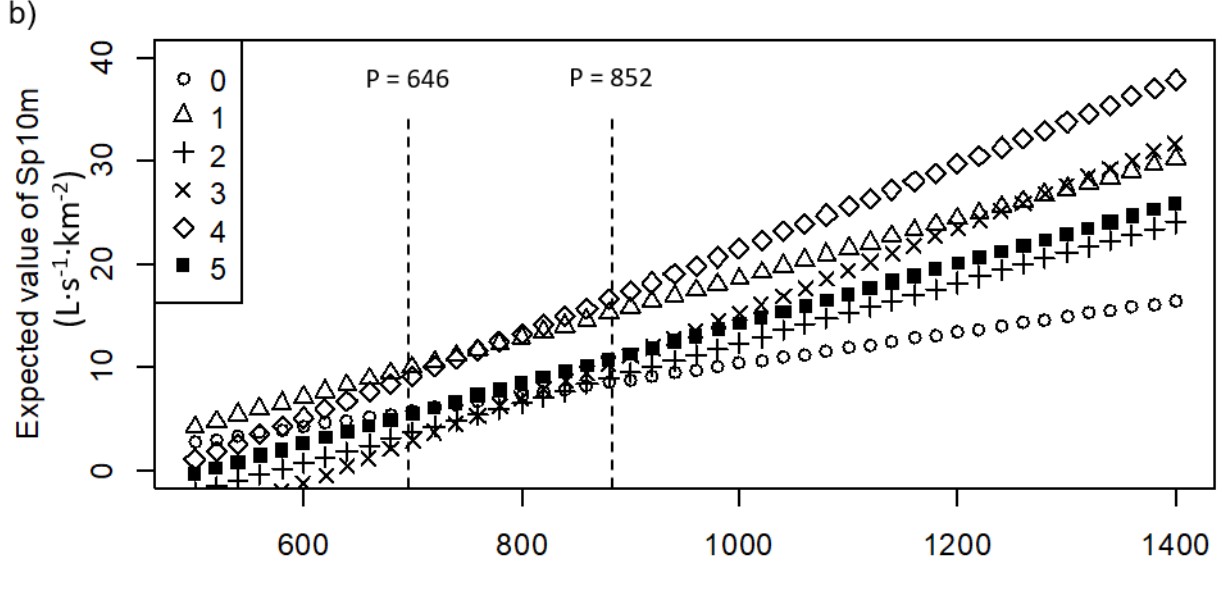





**Appendix A:**

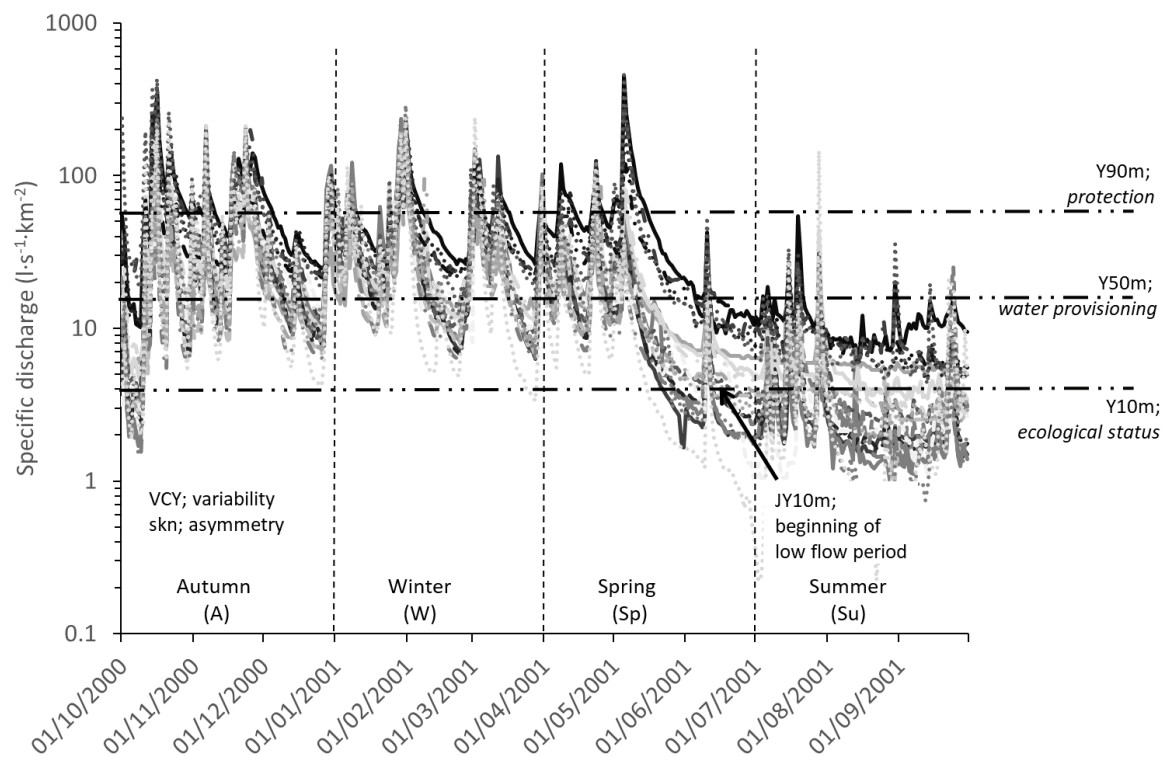





**Appendix B:**







**Appendix C:**

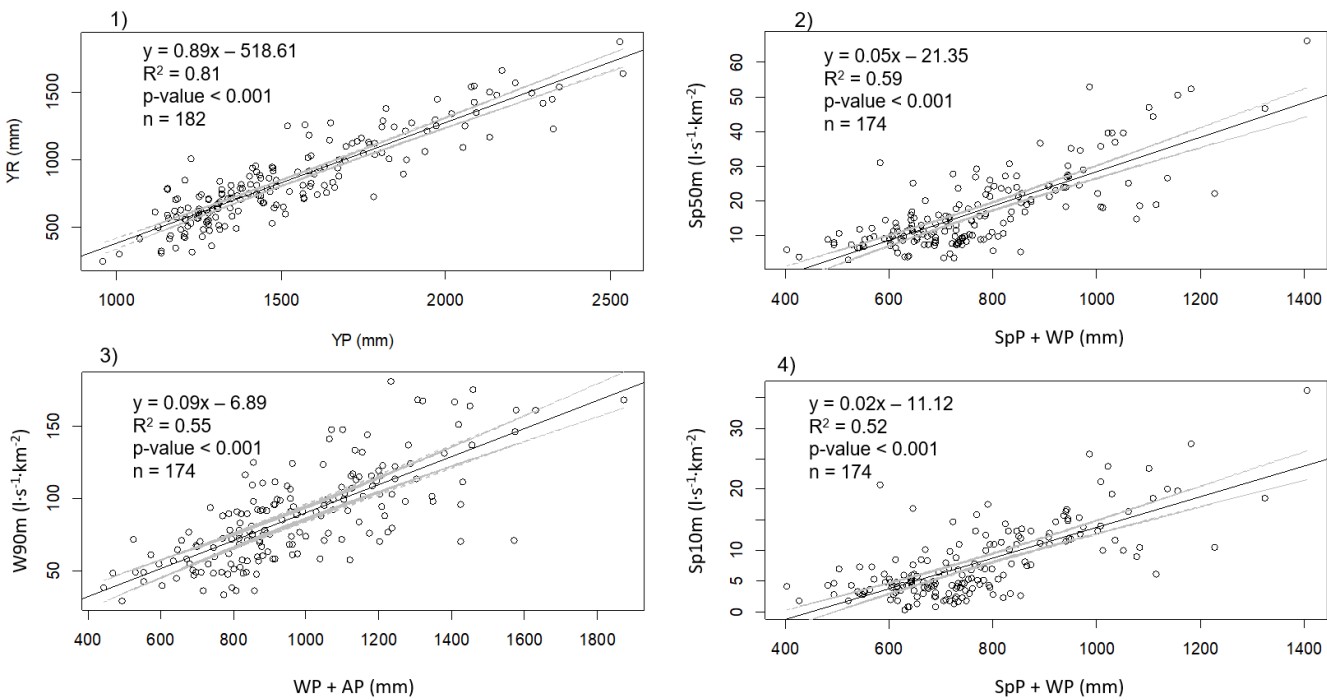



**Appendix D.**

1) Multiple regression model for annual average flows (Y50m) considering alternative land cover and its interaction with annual precipitation (YP).

| D1 | Estimate | Std. Error | t value | p value | Significance |
|---|---|---|---|---|---|
| (Intercept) | 50.67649 | 21.3388 | 2.3749 | 0.01862 | * |
| YP | -0.02049 | 0.0150 | -1.3635 | 0.17446 | |
| Native | -0.80268 | 0.2677 | -2.9985 | 0.00310 | ** |
| Exotic | -0.56718 | 0.2329 | -2.4354 | 0.01586 | * |
| Pasturelands | -0.81356 | 0.2181 | -3.7296 | 0.00026 | *** |
| I(Native * YP) | 0.00046 | 0.0002 | 2.6403 | 0.00902 | ** |
| I(Exotic *YP) | 0.00030 | 0.0002 | 1.8558 | 0.06513 | . |
| I(Pasturelands * YP) | 0.00057 | 0.0002 | 3.4285 | 0.00075 | *** |

Signif. codes:  0 '***' 0.001 '**' 0.01 '*' 0.05 '.' 0.1 ' ' 1

Residual standard error: 3.96 on 178 degrees of freedom
Multiple R-squared: 0.7791,   Adjusted R-squared: 0.7704
F-statistic: 89.69 on 7 and 178 DF,  p-value: $< 2.2e-16$

2) Multiple regression model for average discharge for spring (Sp50m) considering alternative land cover and its interaction with seasonal precipitation (spring + winter precipitation, SpPt).

| | Estimate | Std. Error | t value | p value | Significance |
|---|---|---|---|---|---|
| (Intercept) | 84.01452 | 43.1914 | 1.9452 | 0.05345 | . |
| SpPt (SpP+WP) | -0.06791 | 0.0586 | -1.1585 | 0.24833 | |
| Native | -1.44268 | 0.5367 | -2.6879 | 0.00792 | ** |
| Exotic | -0.86076 | 0.5230 | -1.6457 | 0.10172 | |
| Pasturelands | -0.98659 | 0.3952 | -2.4965 | 0.01352 | * |
| I(Native * SpPt) | 0.00159 | 0.0007 | 2.3249 | 0.02129 | * |
| I(Exotic *SpPt) | 0.00083 | 0.0007 | 1.1590 | 0.24811 | |
| I(Pasturelands * SpPt) | 0.00127 | 0.0006 | 2.2893 | 0.02332 | * |

Signif. codes:  0 '***' 0.001 '**' 0.01 '*' 0.05 '.' 0.1 ' ' 1

Residual standard error: 6.784 on 166 degrees of freedom
Multiple R-squared: 0.6407,   Adjusted R-squared: 0.6256
F-statistic: 42.29 on 7 and 166 DF,  p-value: $< 2.2e-16$

3) Multiple regression model for annual high flows (Y90m) considering alternative land cover and its interaction with annual precipitation (YP).



| | Estimate | Std. Error | t value | p value | Significance |
|---|---|---|---|---|---|
| (Intercept) | -2.86E+02 | 1.06E+02 | -2.707 | 0.007451 | ** |
| YP | 2.46E-01 | 7.31E-02 | 3.364 | 0.000941 | *** |
| Native | 2.88E+00 | 1.14E+00 | 2.5281 | 0.01234 | * |
| Exotic | 2.02E+00 | 1.14E+00 | 1.766 | 0.079105 | . |
| Pasturelands | 3.44E+00 | 1.19E+00 | 2.8874 | 0.004367 | ** |
| I(Native * YP) | -2.07E-03 | 7.54E-04 | -2.7403 | 0.006764 | ** |
| I(Exotic *YP) | -1.54E-03 | 7.85E-04 | -1.9637 | 0.051117 | . |
| I(Pasturelands * YP) | -2.45E-03 | 8.50E-04 | -2.8772 | 0.004503 | ** |

Signif. codes:  0 '***' 0.001 '**' 0.01 '*' 0.05 '.' 0.1 ' ' 1

Residual standard error: 13.5 on 178 degrees of freedom
Multiple R-squared: 0.6837,   Adjusted R-squared: 0.6713
F-statistic: 54.97 on 7 and 178 DF,  p-value: < 2.2e-16

4) Multiple regression model for low flows for spring (Sp10m) considering alternative land cover and its interaction with seasonal precipitation (spring + winter precipitation, SpPt).

| | Estimate | Std. Error | t value | p value | Significance |
|---|---|---|---|---|---|
| (Intercept) | 65.57420 | 33.1084 | 1.9806 | 0.04929 | * |
| SpPt (SpP+WP) | -0.06361 | 0.0458 | -1.3876 | 0.16712 | |
| Native | -0.99263 | 0.3915 | -2.5358 | 0.01214 | * |
| Exotic | -0.65681 | 0.3678 | -1.7858 | 0.07596 | . |
| Pasturelands | -0.80725 | 0.3327 | -2.4266 | 0.01631 | * |
| I(Native * SpPt) | 0.00109 | 0.0005 | 2.1315 | 0.03452 | * |
| I(Exotic *SpPt) | 0.00070 | 0.0005 | 1.3761 | 0.17065 | |
| I(Pasturelands * SpPt) | 0.00108 | 0.0005 | 2.2733 | 0.02429 | * |

Significance codes:  0 '***' 0.001 '**' 0.01 '*' 0.05 '.' 0.1 ' '

Residual standard error: 3.955 on 166 degrees of freedom
Multiple R-squared: 0.5759,   Adjusted R-squared: 0.558
F-statistic: 32.2 on 7 and 166 DF,  p-value: < 2.2e-16

