# Peer review of "Land cover effects on hydrologic services under a precipitation gradient"

_Hydrology and Earth System Sciences, 2018_

## Referee Comment (RC1) · Anonymous Referee #1 · 3 Sep 2018

General comments: The paper "Land cover effects on hydrologic services under a precipitation gradient" describes an exhaustive statistical analysis to evaluate the role of vegetation on the hydrology of a large region with a precipitation gradient in Northern Spain. The authors utilized a large amount of data to test the impacts of forest (native and exotic), as well as pastureland on hydrological variables. The manuscript reads well, is well organized, the objectives are clearly defined, the authors were supported by a good choice of bibliography in the field. It is an interesting paper, however, there are some aspects that require improvement before being considered for publication. In particular, the use of the term "hydrological services" that was not matured enough through the text, and the use of so many tables and appendixes. Specific comments and suggestions are included bellow. I believe they can be addressed with additional,

descriptive text, and no additional analysis are needed.

Specific comments: Page 2, lines 15-20: The title expresses the concept of hydrological services, a term used to specifically relate the benefits people obtain from ecosystems related to water. These include, not only water quantity and regulation but also the dimension of quality. It is well recognized the role of forest on reducing diffuse water pollution. Since the manuscript only focus on the flows of water and their relations to land-use/cover, but never gets the human dimension of the hydrological services, why to use the term hydrological services and not simply water flows. My feeling is that the term "hydrological services" is not sufficiently polished in the Introduction section and throughout the manuscript to be used in the title. I suggest to revise the title accordingly, or to emphasize the concept of hydrological services, not only in the Introduction but throughout the text.

Page 4, line 1-2: Were the oak forests removed for plantations of pinus radiata? 39-48% of the area refers to total area or to forest area?

It is not clear in section 3.4, at what scale were the statistical analysis performed? I supposed for the entire 20 catchments, but then the aggregated value for the entire region masks geographical asymmetries.

The combinations used do not differ substantially from each other, why not to reduce for 3 combinations. Perhaps instead of number, the combinations should have a name, e.g. more exotic, more pasture, in order to better guide the reader in the further tables of results.

It was not clear where do these combinations applied? To the entire catchment. Why to call realistic. They seem to be scenarios of land cover? I recommend to clarify better these methodology for the readers understand results. Page 6: in which software were the statistical analysis carried out? Page 8, lines 7-10: This discussion is well organized and results are well-thought-out in perspective against work done by others. However, please consider that sometimes the growth of forest can have only a slight

effect, depending of course on other environmental factors, such as depth o soil, precipitation episodes, etc. Please consider to take a look on this publication: (Hawtree et al., 2015)

I would like to suggest a table with major findings of the study, a kind of summary to guide the reader. And also add a paragraph on this in the discussion. I think this table will summarize the work and can maybe substitute Table 5 and 6. Are forest fires a threat in this region? Are there any other possible factor that may influence discharge, e.g. forest management that is worthy to discuss? If yes for fire or other factors please add a paragraph in the discussion section around these topics.

Tables 4 to 6 – Did you dived results from catchments with high and low precipitation? Why not the same amount of precipitation in all analysis? Seeing Figures 2 and 3 I got the meaning of this precipitation division. I recommend this to be explained in the captions of the tables and in the methods section. In addition, please put what does each acronym in the table means, at least in the caption, otherwise it is difficult to follow.

I was missing a discussion around the practicability of these findings. Are they useful for policy makers? In which way?

Page 11, Lines 10 to 20: For me these sentences are more a part of discussion. You could create an extra section in the discussion section with major findings and policy implications. I would like to see a conclusion more summarized and without comparing the work with others.

Technical corrections: Page 2, line 18: Keestra et al., 2018, appears "Keesstra" in the References, which one is correct?

Figure 1: The figure is very informative. However, in figure 1 a) it was difficult to understand the border of the catchments, it is difficult to read what are the dashed lines for and the bold lines for. Maybe to delete the dashed lines since they are not

[Figure]

informative and highlight the border of the region and dashed lines the border of the catchments. Update legend accordingly. In the legend, please mention the spatial resolution and source of land cover maps. Table 1: Please detail where are land cover maps 2002 and 2009 coming from, spatial resolution, source. Table 2: I would like to suggest the meaning of the abbreviations described in the table, otherwise the reader has to look for these different meanings spread all over the text. The table as it is not informative. Please also put native and exotic 2 or 3 spaces right, because they are inside of forest section.

Page 5, line 11: where is Fig A1? Please cite Appendix A1.

Page 7, line 11: Fig. C1 is Appendix C? Please use the term Appendix, otherwise people will not understand. If figures in the Appendixes are so important for the study, why not to introduce them in the body of the text? Readers will need to have extra work downloading appendix to fully understand the paper. This is a suggestion, please consider if you feel that all the figures are important.

Please mention the periods of analysis, either in the captions (as Appendix A has), as well in the text (e.g. page 7 line 12 is missing the period of analysis).

Page 7, line 24. Please use "from" instead of "of". Figure 2a shows results "from" the multiple regression

Page 8, line 23: where are the performance statistics shown? Page 8, line 26: in which way increases discharge? Values, Table?? Page 8, line 35: which land cover? Please refine the sentence.

Page 9, line 15: Please consider this article on these matters: (Carrick et al., 2018)Carrick, J., Bin Abdul Rahim, M.S.A., Adjei, C., Ashraa Kalee, H.H.H., Banks, S.J., Bolam, F.C., Campos Luna, I.M., Clark, B., Cowton, J., Domingos, I.F.N., Golicha, D.D., Gupta, G., Grainger, M., Hasanaliyeva, G., Hodgson, D.J., Lopez-Capel, E., Magistrali, A.J., Merrell, I.G., Oikeh, I., Othman, M.S., Ranathunga Mudiyanselage, T.K.R.,

Samuel, C.W.C., Sufar, E.K., Watson, P.A., Zakaria, N.N.A.B., Stewart, G., 2018. Is Planting Trees the Solution to Reducing Flood Risks? J. Flood Risk Manag. 1–10. https://doi.org/10.1111/jfr3.12484 Page 11, line 23. Please remove "in" in the beginning of the sentence. Page 11, line 27: There simply is no unique "best combination. Please remove "simply".

---

## Referee Comment (RC2) · Anonymous Referee #2 · 7 Sep 2018

This paper, discussing the influence of precipitation and land cover on hydrological indicators, is well-written and fits well within the scope of HESS. There are a few things to clarify though before publication, as listed below. The main things I would like to see more clarified are how the base and 5 other land cover combinations are created and how dependent your conclusions are on changes in your assumptions (for instance a slight change in those combinations, or using only seasonal instead of 6 month precipitation, etc).

Introduction: P2, line 8-10: what is the deforestation in hectares and /or the afforestation in %?

Study area: P3: is the study area a closed drainage network (I assume so) or is there inflow from other / higher regions?

Methodology: P4, line 26-30: for clarity you could already indicate here that the land cover for 2002 and 2009 is very similar, to support the 'merging' of the two 5-year periods of hydrologic observations P4: line 31: in specific discharge unit L/s/km2 what is L? The letter would indicate a length but specific discharge = discharge / area so L would be a volume? P5, line 29-20: 'Seasonal precipitation amounts . . . were also computed', are they also based on estimates from the Environment and Hydraulic Works Department like annual precipitation or do you yourself compute them from the annual precipitation? P6, line 21-34: this section would be a bit more clear if you switch the paragraph starting in line 30 with that starting in line 25, since the "real values of explanatory variables" in line 23 are the land cover combinations explained starting from line 30 if I understand correctly? P6 line 25-29: How sensitive are your results to the choice to exclude outliers and to add precipitation of the previous study? i.e. are your conclusions different if you do not exclude outliers and / or only include precipitation of the 3-month season? P6 line 30 – P7 line 5: are your 6 land cover combinations the most common ones in the region? If so how much of the area do they represent?

Results and discussion: P8 line 20: Table 6 is mentioned before Table 5. Furthermore, you could mention in the table captions that you do not show insignificant results (now I wondered for instance why Table 6 did not include Sp90m – that results for Sp90m are insignificant is only mentioned in later sections). P8 line 21: I think you refer to figure 2b (Sp50m) instead if 3b (Sp10m) P9 line 14-16: what potentials are usually claimed?

---

## Author Comment (AC1) · 14 Sep 2018

First of all the authors want to thank the Referee 1 for his very helpful comments which are contributing, with no doubt, to the improvement of the manuscript. In the following we address the comments. All the changes included in the manscript itself, as well as in the tables and figures are included in the suplement file.

Anonymous Referee #1 General comments: The paper "Land cover effects on hydrologic services under a precipitation gradient" describes an exhaustive statistical analysis to evaluate the role of vegetation on the hydrology of a large region with a precipitation gradient in Northern Spain. The authors utilized a large amount of data to test the impacts of forest (native and exotic), as well as pastureland on hydrological

variables. The manuscript reads well, is well organized, the objectives are clearly defined, the authors were supported by a good choice of bibliography in the field. It is an interesting paper, however, there are some aspects that require improvement before being considered for publication. In particular, the use of the term "hydrological services" that was not matured enough through the text, and the use of so many tables and appendixes. Specific comments and suggestions are included bellow. I believe they can be addressed with additional, descriptive text, and no additional analysis are needed.

Specific comments: RC 1: Page 2, lines 15-20: The title expresses the concept of hydrological services, a term used to specifically relate the benefits people obtain from ecosystems related to water. These include, not only water quantity and regulation but also the dimension of quality. It is well recognized the role of forest on reducing diffuse water pollution. Since the manuscript only focus on the flows of water and their relations to land-use/cover, but never gets the human dimension of the hydrological services, why to use the term hydrological services and not simply water flows. My feeling is that the term "hydrological services" is not sufficiently polished in the Introduction section and throughout the manuscript to be used in the title. I suggest to revise the title accordingly, or to emphasize the concept of hydrological services, not only in the Introduction but throughout the text.

AC 1: All the text, specially the results and discussion section and the conclusions were reviewed in this sense. Water flows were linked to hydrological services more clearly. In this sense, terms as water provisioning, flood protection or conservation of ecological status, have been used when referring to average, high or low flows along the text. Additionally, as later suggested by this reviewer, a new subsection was included at the end the results section, highlighting this dimension of the work.

P11; L18: "As Ellison et al. (2017) stated, the impact of land management politics on hydrological services is not usually considered, however, taking into account local findings on the relationship between land cover and water-related ecosystem services

is necessary for an adequate integrated catchment management. Results observed in Table 4 and Figures 2 and 3, are in this sense useful to be considered when planning land management, in order to have some knowledge on different trends on hydrologic services that can be derived from different decisions under areas with different precipitation amounts. There is no unique "best combination" for all locations and all services, water provision, flood risk protection, ecological status conservation. . .."

RC 2: Page 4, line 1-2: Were the oak forests removed for plantations of pinus radiata? 39- 48% of the area refers to total area or to forest area?

AC 2: In some cases the introduction of exotic species occurred directly removing broad-leaved forest to plant Pinus radia. In other cases, the native forest was first removed to create spaces for pasture, that later, were abandoned to plant exotic species. 39- 48% of the area refers to the potential oak forest area.

The orginal text at this point: "These exotic species were introduced in the second half of the twentieth century as a result of government support for afforestation policies and currently cover 39–48 % of the area that could sustain oak forests (Garmendia et al., 2012). The abandonment of traditional cattle and sheep farming practices has also contributed to the conversion of pastureland and rangelands to fast-growth exotic plantations (Ruiz Urrestarazu, 1999)."

was modified as follows trying to clarify these points: P4; L1: "These exotic species were introduced in the second half of the twentieth century as a result of government support for afforestation policies. The abandonment of traditional cattle and sheep farming practices has also contributed to the conversion of pastureland and rangelands, most of them previously converted from broad-leaved forests, to fast-growth exotic plantations (Ruiz Urrestarazu, 1999). As a result, those exotic species currently cover 39–48 % of the potential oak forests (Garmendia et al., 2012)."

RC 3: It is not clear in section 3.4, at what scale were the statistical analysis performed? I supposed for the entire 20 catchments, but then the aggregated value for the entire

region masks geographical asymmetries.

AC 3: As the referee comments the statistical analysis was performed for the 20 catchments. We are not sure of which is the referees concern about geographical asymmetry In fact, precipitation is the variable that shows a higher variability in the area, and that is the reason why we considered it necessary to be included in the analysis.

In order to clarify this point in the manuscript, the original text: "As a first step, the influence of precipitation on different hydrological indices was analysed using the following simple linear regression, Eq. (1)"

Was modified as follows: P6; L11: "As a first step, the influence of precipitation on different hydrological indices was analysed using the following simple linear regression that included the 20 catchment, Eq. (1):"

RC 4: The combinations used do not differ substantially from each other, why not to reduce for 3 combinations. Perhaps instead of number, the combinations should have a name, e.g. more exotic, more pasture, in order to better guide the reader in the further tables of results. It was not clear where do these combinations applied? To the entire catchment. Why to call realistic. They seem to be scenarios of land cover? I recommend to clarify better these methodology for the readers understand results.

AC 4: With the six combinations included, apart from considering maximum of one or of other land use, different average values of exotic, native and pasturelands are considered. In our opinion this helps understanding what may happen, not only when land cover is very different, but how smaller differences can affect to hydrological services. In that sense, we consider that none of the combinations should be remover. However, and following the recommendation of this reviewer we added in the manuscript a description and an acronym for each of the combinations. These acronyms are also included in table 3 and have been used when describing results. In our opinion, as the referee says, this will better guide the reader through the manuscript.

The combinations were considered for the entire catchment and we call them realistic because they are built using real data obtained from table 3 (this point was also clarified in the manuscript). We decided to call them combinations instead of scenarios to avoid confusion with other studies based on hydrological modelling, and because land cover alternatives are not considered, as in modelling, for certain areas in the catchment but globally in the regression equation.

In order to include the reviewer proposals on this topic in the manuscript, the original text: "The different land cover combinations shown in Table 3 were explored under low and high precipitation conditions, and compared to a "base" land cover combination (combination 0) of 76 % exotic, 18 % pastureland, and 6 % native. Land cover combination 0 was defined as a combination with a maximum area of exotic plantations, minimum area of native forests, and a low percentage of pasturelands (calculated as the remaining percentage to cover 100 % of the area). Combinations from 1 to 5 were defined as realistic alternative patterns to combination 0. Differences between these patterns and combination 0 were calculated for each hydrological index under low and high precipitation conditions (Tables 4, 5 and 6). These combinations were defined considering real data (e.g., maximum, minimum, or mean percentages of native forests, exotic plantations, and pasturelands). Defined in this way, each combination was used to examine interactions between realistic data; results for scenarios that might be very different from the existing ones were not extrapolated."

Was modified as follows: P7; L3: "The different land cover combinations shown in Table 3 were explored for the catchments, under low and high precipitation conditions, and compared to a "base" land cover combination (combination EXO) of 76 % exotic, 18 % pastureland, and 6 % native. Land cover combination EXO was defined as a combination with a maximum area of exotic plantations, minimum area of native forests, and a low percentage of pasturelands (calculated as the remaining percentage to cover 100 % of the area). Other 5 combinations were defined as realistic alternative patterns to combination EXO as they were calculated considering real data (e.g., maximum,

minimum, or mean percentages of native forests (NAT), exotic plantations (EXO), and pasturelands (PAST) (see Table 2)) and considering the sum of the three as 100 %. Following this approach combination EXO + PAST represents high percentages of exotic plantations and pastureland, combination EXO + NAT high percentage of forest, combination NAT high percentage of native forest, combination NAT + PAST is mostly native forests and pasturelands and combination EXO + NAT + PAST a mixture of average percentages of exotic plantations, native forests and pasturelands. Differences between these patterns and combination EXO were calculated for each hydrological index under low and high precipitation conditions (Table 4). Defined in this way, each combination was used to examine interactions between realistic data; results for combinations that might be very different from the existing ones were not extrapolated."

RC 5: Page 6: in which software were the statistical analysis carried out?

AC 5: All the statistical analysis was programmed and performed using the free software R in its version 3.1.2 in the R studio interface.

The following text was included in point 3.4 of the manuscript: P6; L27: "All the statistical analysis were programmed and performed using the free software R in its version 3.1.2 in the R studio interface."

RC 6: Page 8, lines 7-10: This discussion is well organized and results are well-thought-out in perspective against work done by others. However, please consider that sometimes the growth of forest can have only a slight effect, depending of course on other environmental factors, such as depth of soil, precipitation episodes, etc. Please consider to take a look on this publication: (Hawtree et al., 2015)

AC 6: This reference was considered in the new version of the manuscript as follows: P10; L32: "No statistically significant influences were observed on median, high, or low autumn and summer flows or on other hydrological indices related to the timing of low flows or other changes to the hydrograph; however, this does not rule out the possibility of relationships between land cover and the hydrological indices. As shown

in Appendix C, precipitation (volume and distribution) is the main driver of the system, and thus the influence of other drivers, such as land cover, may fail to emerge as statistically significant. Additionally, there may be other environmental factors, such as soil depth, as Hawtree et al., (2015) found in a catchment of north-central Portugal."

RC 7: I would like to suggest a table with major findings of the study, a kind of summary to guide the reader. And also add a paragraph on this in the discussion. I think this table will summarize the work and can maybe substitute Table 5 and 6.

AC 7: Following the suggestions of reviewer 1 all results included in previous table 4, 5 and 6 were merged in one table that includes all findings (Table 4 in the reviewed version of the manuscript). And a global discussion, previously included in conclusions, is now included in the results section as follows: P11; L6: "Clear conclusions about the effect of each land cover combination on hydrological services cannot be drawn without considering the amount of precipitation. However, results show that in the Bay of Biscay area, the presence of any kind of forest decreases water provisioning service (Y50m), and this effect is more evident with exotic plantations as the annual precipitation increases. Additionally, similar to other studies (Robinson et al., 2003; Carrick et al., 2017), this study indicates that the potential for forests to reduce flooding risk is low; however, the effect of land cover on high flows also changes with precipitation. For low precipitation amounts, forests, especially exotic plantations, show greater potential to reduce annual and wintertime high flows than pasturelands, but this potential decreases as annual or seasonal precipitation increases. Moreover, when high annual precipitation is considered, the potential of exotic plantations to reduce flood magnitude is lower than that of native forests or pasturelands. Further, the results also show that exotic plantations have a slight positive effect on annual low flows under low annual precipitation conditions; however, low flows increase as annual precipitation increases and when exotic forests are replaced by pasturelands. This effect is most evident in winter and spring, and when the combination of pasturelands and native forests account for most of the catchment area."

RC 8: Are forest fires a threat in this region? Are there any other possible factor that may influence discharge, e.g. forest management that is worthy to discuss? If yes for fire or other factors please add a paragraph in the discussion section around these topics.

AC 8: Forest fires are not a main threat in this region, at least not as in other regions of the Iberian Peninsula. Forest management is an important issue, specially that related to exotic species. In fact, a text was already included explaining the specificity of those plantations in the study area. However, those are the current practices, and there are not, at least at significant spatial scales, other type of managements that could imply hydrological differences between catchments with exotic plantation. The text we refer to says as follows:

P11; L28: "Further study is needed in the Bay of Biscay area to determine how the characteristics of specific tree species (e.g., their phenology and physiology) affect various components of the hydrological cycle. Analyses are also needed to establish the relationship between forest types, land management issues and soil development. For instance, clearcutting of exotic species in the study area is usually accompanied by harvesting with chainsaws, skidding, and mechanical site preparation (prior to re-planting) such as scarification and ripping (Gartzia-Bengoetexea et al., 2009). These logging operations alter the physical properties of soil, affecting processes such as infiltration, evapotranspiration, percolation, and lateral flow, and in turn, catchment water balance and temporal distribution river discharge. A deepened understanding of those relationships will help achieve a solid understanding of how trees characteristics, forest types and management strategies, and soil properties influence water flows."

RC 9: Tables 4 to 6 – Did you dived results from catchments with high and low precipitation? Why not the same amount of precipitation in all analysis? Seeing Figures 2 and 3 I got the meaning of this precipitation division. I recommend this to be explained in the captions of the tables and in the methods section. In addition, please put what does each acronym in the table means, at least in the caption, otherwise it is difficult to

follow.

RC 9: As the referee already guessed, there is no catchment division in the analysis, but the same analysis was repeated with low and high precipitation amounts. As it is already explained in the methodology section, P6; L29: "The objective of this study was to compare predicted hydrological indices for various land cover combinations under different precipitation amounts. To avoid biased results affected by considering extreme values, the 1st and 3rd quartiles of the precipitation data series were calculated for the selected period (annual or seasonal) and defined as the low and high precipitation conditions." …... P7; L3: "The different land cover combinations shown in Table 3 were explored for the catchments, under low and high precipitation conditions, and compared to a "base" land cover combination". However, and reading the question of the referee, we realize that this idea needs to be reinforced and, following the suggestion of the referee we include some explanation in the caption of table 4.

The original caption: "Table 4. Results obtained, from multiple regression models, in the variation in percentage with respect to the base land cover combination (EXO) in the annual median, high and low flows with land cover for low and high precipitation amounts."

Was modified as follows (table 4 as example): "Table 4. Results obtained, from multiple regression models, for base land cover combination (EXO) and for alternative combinations defined in Table 3 expressed as the difference (in %) with respect to EXO combination. Results for annual, winter and spring median, high and low flows are included, for low (1st quartile) and high (3rd quartile) precipitation amounts (see Figs. 2 and 3). Y50m, Y90m and Y10m refer to median, high and low flows of the annual discharge series; W90m and W10m refer to high and low flows of the winter discharge series; and Sp50m and Sp10m refer to median and low flows of the spring discharge series."

RC 10: I was missing a discussion around the practicability of these findings. Are they

useful for policy makers? In which way?

RC 10: A text on this was included in the new subsection of the manuscript: "4.4 Land cover effects on hydrological services. Implications."

P11; L18: "As Ellison et al. (2017) stated, the impact of land management politics on hydrological services is not usually considered, however, taking into account local findings on the relationship between land cover and water-related ecosystem services is necessary for an adequate integrated catchment management. Results observed in Table 4 and Figures 2 and 3, are in this sense useful to be considered when planning land management, in order to have some knowledge on different trends on hydrologic services that can be derived from different decisions under areas with different precipitation amounts. There is no unique "best combination" for all locations and all services, water provision, flood risk protection, ecological status conservation. However, the effect of different land cover combinations, apart from those analysed in this paper, and always inside the limits those included in the multiple regression models proposed, on different hydrological services may be applied. Results obtained, should be in the range of those shown in figures 2 and 3, and could be used to compare the benefits and disadvantages in each of the commented services."

RC 11: Page 11, Lines 10 to 20: For me these sentences are more a part of discussion. You could create an extra section in the discussion section with major findings and policy implications. I would like to see a conclusion more summarized and without comparing the work with others.

RC 11: These finding were moved to the new subsection of the manuscript: "4.5 Land cover effects on hydrological services. Implications."

Technical corrections: RC 12: Page 2, line 18: Keestra et al., 2018, appears "Keesstra" in the References, which one is correct?

AC 12: We are sorry for the mistake. Keestra is the correct form. It has been corrected

in the reference list of the new version of the manuscript

RC 13: Figure 1: The figure is very informative. However, in figure 1 a) it was difficult to understand the border of the catchments, it is difficult to read what are the dashed lines for and the bold lines for. Maybe to delete the dashed lines since they are not informative and highlight the border of the region and dashed lines the border of the catchments. Update legend accordingly. In the legend, please mention the spatial resolution and source of land cover maps.

AC 13: Figure 1 was modified following the recommendations of this reviewer and trying to make it clearer (see new figure in the reviewed version of the manuscript). Legend was updated and spatial resolution and source of maps were included in the figure caption.

The original caption: "Figure 1. a) Location and digital terrain model of the study area with the main drainage network, location of gauging stations, catchments, and average precipitation values. b) Land cover map of the study area in 2002 and 2009."

Was modified as follows: "Figure 1. a) Location and digital terrain model (5x5 m of resolution) of the study area with the main drainage network, location of gauging stations, catchments, and average precipitation values. b) Land cover map of the study area in 2002 (IFN3, 2005) and 2009 (IFN4, 2011) at the 1:25000 scale."

RC 14: Table 1: Please detail where are land cover maps 2002 and 2009 coming from, spatial resolution, source.

AC 14: Spatial resolution and source of maps were included in the table caption. Some more information was also included in the caption.

The original caption: "Table 1. Catchment descriptions." Was modified as follows: "Table 1. Catchment descriptions. Code of the gauging station, catchment name, catchment area, primary land cover types percentages for 2002 (IFN3, 2005) and 2009 (IFN4, 2011) at the 1:25000 scale."

[Figure]

RC 15: Table 2: I would like to suggest the meaning of the abbreviations described in the table, otherwise the reader has to look for these different meanings spread all over the text. The table as it is not informative. Please also put native and exotic 2 or 3 spaces right, because they are inside of forest section.

AC 15: A footnote was included in the table with abbreviation meaning. Table caption was modified in accordance with this change.

The original caption: "Table 2. Statistics of hydrological indices, land cover types, and precipitation amounts used in this study. See the text for abbreviation meaning."

Was modified as follows: "Table 2. Statistics of hydrological indices, land cover types, and precipitation amounts used in this study."

Add footnote was included: "* YR: annual runoff; Y10m, A10m, W10m, Sp10m and Su10m: 10th percentile for annual, autumn, winter, spring and summer discharge series; Y50m, A50m, W50m, Sp50m and Su50m: 50th percentile for annual, autumn, winter, spring and summer discharge series; Y90m, A90m, W90m, Sp90m and Su90m: 90th percentile for annual, autumn, winter, spring and summer discharge series; CVY, CVA, CVW, CVSp and CVSu: coefficient of variation for annual, autumn, winter, spring and summer discharge series; skn: skewnwss of the annual discharge series; JY10m: Julian day of the beginning of the low flow period. YP: annual precipitation amount for the catchment; AP + SuP: summer plus previous autumn precipitation amount for the catchment; WP + AP: winter plus previous autumn precipitation amount for the catchment; SpP + WP: spring plus previous winter precipitation amount for the catchment; SuP + SpP: summer plus previous spring precipitation amount for the catchment. See the text for more information."

RC 16: Page 5, line 11: where is Fig A1? Please cite Appendix A1.

AC 16: The text was reviewed accordingly.

Where it said: "(Fig. A1)"

Now it says: "(Appendix A)"

RC 17: Page 7, line 11: Fig. C1 is Appendix C? Please use the term Appendix, otherwise people will not understand.

AC 17: The text was reviewed accordingly.

Where it said: "(Fig. C1)"

Now it says: "(Appendix C)"

RC 18: If figures in the Appendixes are so important for the study, why not to introduce them in the body of the text? Readers will need to have extra work downloading appendix to fully understand the paper. This is a suggestion, please consider if you feel that all the figures are important.

AC 18: In our opinion appendixes help going further in the context of the paper, however, they are not very necessary to understand the main findings of the paper. Introducing them in the text would mean too many figures and tables and as HESS gives authors the chance to include some extra material as appendices we thought to take advantage of it, in order to extend information for those readers more interested in the hydro-meteorological context or the more specifically statistical results.

RC 19: Please mention the periods of analysis, either in the captions (as Appendix A has), as well in the text (e.g. page 7 line 12 is missing the period of analysis).

AC 19: Periods of analysis were included in the text and in the caption of table 2.

Original text in the manuscript: "The regression includes all data collected during the study period ..."

Was modified as follows: P7; L21: "The regression includes all data collected during the study period (from 2000–2001 to 2004–2005 and from 2007–2008 to 2011–2012)"

RC 20: Page 7, line 24. Please use "from" instead of "of". Figure 2a shows results
"from" the multiple regression

Suggested change was made.

RC 21: Page 8, line 23: where are the performance statistics shown?

All the performance statistics are included in Appendix D. The text was modified for clarification at this point. However, Figures 2 and 3 were also modified in order to include some statistics and make the understanding of the text easier. See modified figure in the new version of the manuscript.

The original text: "Regression results shown in Fig. 3b indicate that native forests, pasturelands, and precipitation in interactions with both types of land cover are significant (p values <0.05). The percentage of exotic plantations in the catchment significantly influences the median discharge in the spring. The coefficient of determination for this graph is 0.63, indicating a good fit."

Was modified as follows: P8; L33: "Regression in Fig. 2b show significant effect of native forests, pasturelands, and precipitation in interactions with both types of land cover (p values <0.05) on Sp50m. The percentage of exotic plantations in the catchment significantly influences the median discharge in the spring. The coefficient of determination for this graph is 0.63, indicating a good fit."

RC 22: Page 8, line 26: in which way increases discharge? Values, Table??

AC 22: The text was slightly modified at this point to include some data and table reference.

The original text: "Decreasing the percentage of exotic plantations (by increasing pasturelands or native forests) increases spring discharge under high precipitation amounts (Sp50m). The magnitude of this increase is larger when the extension of pasturelands is larger (combinations 1 (EXO) and 4 (NAT+PAST)). At low precipitation rates, the positive effect of pasturelands on Sp50m remains, while increasing native forest (combinations 2 (EXO+NAT) and 3 (NAT)) has a negative effect on median discharge."

Was modified as follows: P8; L3: "Decreasing the percentage of exotic plantations (by increasing pasturelands or native forests) increases spring average discharge (Sp50m), up to a 76 %, under high precipitation amounts (852 mm) (Table 4). The magnitude of this increase is larger when the extension of pasturelands is larger (combinations EXO and NAT+PAST). At low precipitation rates (646 mm), the positive effect of pasturelands on Sp50m remains, while increasing native forest (combinations EXO+NAT and NAT) has a negative effect on median discharge."

RC 23: Page 8, line 35: which land cover? Please refine the sentence.

AC 23: Authors do not understand this question. Is the referee referring to next sentences? P9; L11: "Conversely, land cover combinations 3 and 4 (with the highest percentage of native forests and pasturelands, respectively, and lowest percentage of exotic plantations) show the highest variation in Y50m and Sp50m in the precipitation gradient existing in the study area. Consequently, the land cover combination that gives the highest or lowest median discharge values changes with annual and seasonal precipitation amounts."

RC 24: Page 9, line 15: Please consider this article on these matters: (Carrick et al., 2018)Carrick, J., Bin Abdul Rahim, M.S.A., Adjei, C., Ashraa Kalee, H.H.H., Banks, S.J., Bolam, F.C., Campos Luna, I.M., Clark, B., Cowton, J., Domingos, I.F.N., Golicha, D.D., Gupta, G., Grainger, M., Hasanaliyeva, G., Hodgson, D.J., Lopez-Capel, E., Magistrali, A.J., Merrell, I.G., Oikeh, I., Othman, M.S., Ranathunga Mudiyanselage, T.K.R., Samuel, C.W.C., Sufar, E.K., Watson, P.A., Zakaria, N.N.A.B., Stewart, G., 2018. Is Planting Trees the Solution to Reducing Flood Risks? J. Flood Risk Manag. 1–10. https://doi.org/10.1111/jfr3.12484

AC 24: A reference to this paper was included in the results section of the new version of the manuscript.

[Figure]

P10; L5: "In this sense Carrick et al., (2017) after a met-analysis of 156 papers, concluded that a weak direct evidence of the effects of tree cover on flood risk, due to the high uncertainty found in results."

RC 25: Page 11, line 23. Please remove "in" in the beginning of the sentence.

AC 25: Sorry for this error. "in" was removed from the sentence.

RC 26: Page 11, line 27: There simply is no unique "best combination. Please remove "simply".

AC 26: Suggested change was made.

Please also note the supplement to this comment:
https://www.hydrol-earth-syst-sci-discuss.net/hess-2018-366/hess-2018-366-AC1-supplement.pdf

**Supplement:**

[revised manuscript text omitted]

---

## Author Comment (AC2) · 14 Sep 2018

First of all the authors want to thank the Referee 2 for his comments which are contributing to the improvement of the manuscript. In the following we address the comments. All the changes included in the manscript itself, as well as in the tables and figures are included in the suplement file.

Anonymous Referee #2 This paper, discussing the influence of precipitation and land cover on hydrological indicators, is well-written and fits well within the scope of HESS. There are a few things to clarify though before publication, as listed below.

RC 1: The main things I would like to see more clarified are how the base and 5 other land cover combinations are created

AC 1: Land cover combination 0 was defined as a combination with a maximum area of exotic plantations, minimum area of native forests, and a low percentage of pasturelands (calculated as the remaining percentage to cover 100 % of the area). Combinations from 1 to 5 were calculated considering real data (e.g., maximum, minimum, or mean percentages of native forests, exotic plantations, and pasturelands (see Table 2)) and considering the sum of the three as 100 %.

Following the recommendation of both reviewers we better explained this part of the methodology, and the original text: "The different land cover combinations shown in Table 3 were explored under low and high precipitation conditions, and compared to a "base" land cover combination (combination 0) of 76 % exotic, 18 % pastureland, and 6 % native. Land cover combination 0 was defined as a combination with a maximum area of exotic plantations, minimum area of native forests, and a low percentage of pasturelands (calculated as the remaining percentage to cover 100 % of the area). Combinations from 1 to 5 were defined as realistic alternative patterns to combination 0. Differences between these patterns and combination 0 were calculated for each hydrological index under low and high precipitation conditions (Tables 4, 5 and 6). These combinations were defined considering real data (e.g., maximum, minimum, or mean percentages of native forests, exotic plantations, and pasturelands). Defined in this way, each combination was used to examine interactions between realistic data; results for scenarios that might be very different from the existing ones were not extrapolated."

was modified as follows: P7; L2: "The different land cover combinations shown in Table 3 were explored for the catchments, under low and high precipitation conditions, and compared to a "base" land cover combination (combination EXO) of 76 % exotic, 18 % pastureland, and 6 % native. Land cover combination EXO was defined as a combination with a maximum area of exotic plantations, minimum area of native forests, and a low percentage of pasturelands (calculated as the remaining percentage to cover 100 % of the area). Other 5 combinations were defined as realistic alternative patterns to combination EXO as they were calculated considering real data (e.g., maximum,

minimum, or mean percentages of native forests (NAT), exotic plantations (EXO), and pasturelands (PAST) (see Table 2)) and considering the sum of the three as 100 %. Following this approach combination EXO + PAST represents high percentages of exotic plantations and pastureland, combination EXO + NAT high percentage of forest, combination NAT high percentage of native forest, combination NAT + PAST is mostly native forests and pasturelands and combination EXO + NAT + PAST a mixture of average percentages of exotic plantations, native forests and pasturelands. Differences between these patterns and combination EXO were calculated for each hydrological index under low and high precipitation conditions (Table 4). Defined in this way, each combination was used to examine interactions between realistic data; results for combinations that might be very different from the existing ones were not extrapolated."

RC 2: and how dependent your conclusions are on changes in your assumptions (for instance a slight change in those combinations, or using only seasonal instead of 6 month precipitation, etc).

AC 2: Figures 2 and 3 show the effect of different land cover combinations in hydrological indices considering a precipitation gradient. In this sense, it could be considered that slight changes in those land cover combinations, inside the limits of maximum and minimum cover percentage considered for each of them, should give results inside the range of the obtained lines. However, changing land cover percentages out of the real limits could lead to erroneous extrapolation of results. In any case, this type of analysis should be more considered to obtain trends of changes than to obtain numerical absolute results.

Some new text explaining this was included in the new 4.5 subsection: P11; L24: "...the effect of different land cover combinations, apart from those analysed in this paper, and always inside the limits those included in the multiple regression models proposed, on different hydrological services may be applied. Results obtained, should be in the range of those shown in figures 2 and 3, and could be used to compare the benefits and disadvantages in each of the commented services."

Including precipitation of only one season in the analysis does not give significant statistical results. in order to clarify this, in the original text: "..., while for seasonal scale, precipitation of the season studied plus that of the previous season (6 months total) were considered."

The following was added: P6; L33: "..., while for seasonal scale, precipitation of the season studied plus that of the previous season (6 months total) were considered. The statistical analysis was also carried out considering precipitation of the studied season (3 months), however, no statistically significant results were found."

Introduction: RC 3: P2, line 8-10: what is the deforestation in hectares and /or the afforestation in %?

AC 3: Deforestation between 1990 and 2015 reported in the Global Forest Resources Assessment published by FAO in 2016 was about 129 millions of hectares (FAO, 2016).

This data was included in the original text, where it said: "Worldwide, deforestation rates outstrip afforestation by several million hectares per year. Overall global forest cover declined by 3.25% between 1990 and 2015 (FAO, 2016)"

Now it says: P2; L8: "Worldwide, deforestation rates outstrip afforestation by several million hectares per year. Overall global forest cover declined by 3.25% (129 million ha) between 1990 and 2015 (FAO, 2016)"

FAO: Global Forest Resources Assessment 2015: How Are the World's Forests Changing? 2nd Ed., Food and Agriculture Organization of the United Nations, Rome, Italy, 2016.

Study area: RC 4: P3: is the study area a closed drainage network (I assume so) or is there inflow from other / higher regions?

The study area are 20 catchments located in the Gipuzkoa province with no inflow from other areas. In the manuscript, a first general description of the province is included in order to give some general geo-environmental characterization of the area and after,

specific land cover characteristics of the 20 catchments are resumed in Table 1 and Figure 1.

The text in the manuscript was slightly modified in order to clarify this point: Where it said: "The study area is in Gipuzkoa Province (1980 km2) in the Basque Country..." Now it says: P3; L14: "The studied catchments are located in Gipuzkoa Province (1980 km2), in the Basque Country..."

And where it said: P4; L10: "The study catchments exhibit a diverse mix of land cover types..." Now it says: "The catchments studied in this area exhibit a diverse mix of land cover types..."

Methodology: RC 5: P4, line 26-30: for clarity you could already indicate here that the land cover for 2002 and 2009 is very similar, to support the 'merging' of the two 5-year periods of hydrologic observations

AC 5: In fact, the authors created a unique database that includes both five-year periods, hydrological data are not merged considering one unique land cover distribution. However, land cover data corresponding to hydrological data from 2000-2001 to 2004-2005 is that from the 2002 inventory, and land cover data from the 2009 corresponds to hydrological data from 2007-2008 to 2011-2012.

Original text in the manuscript: "To maintain coherence with land cover data obtained from forest inventories carried out during 2002 and 2009, discharge data was considered for two five-hydrological-year periods. The first period, from 2000–2001 to 2004–2005, was compared with land cover data obtained during 2002 (IFN3, 2005). The second period, from 2007–2008 to 2011–2012, was compared with land cover data from 2009 (IFN4, 2011). In this way, two sets of discharge series, accounting for a total of 10 hydrological-years, were selected for each gauging station. To facilitate comparison among catchment responses, all discharge data, including those for hydrological indicators, are referred to as specific discharges (L s–1 km–2)."

[Figure]

Was slightly modified in order to clarify this point: P4; L28: "To maintain coherence with land cover data obtained from forest inventories carried out during 2002 and 2009, discharge data was considered for two five-hydrological-year periods. Data from the first period, from 2000–2001 to 2004–2005, was compared with land cover data obtained during 2002 (IFN3, 2005). Data from the second period, from 2007–2008 to 2011–2012, was compared with land cover data from 2009 (IFN4, 2011). In this way, hydrological data accounting for 10 hydrological-years were considered for each gauging station. To facilitate comparison among catchment responses, all discharge data, including those for hydrological indicators, are referred to as specific discharges (L s–1 km–2)."

RC 6: P4: line 31: in specific discharge unit L/s/km2 what is L? The letter would indicate a length but specific discharge = discharge / area so L would be a volume?

AC 6: In this case, L refers to a volume, litre, allowed in the SI (international system) and in the manuscript preparation guidelines of HESS. It could lead to confusion in a context in which dimensions are expressed, however, in this case, units are expressed for all dimensions (seconds for time or km2 for area) so that, we do not think it need any clarification in the text.

RC 7: P5, line 29-20: 'Seasonal precipitation amounts : : : were also computed', are they also based on estimates from the Environment and Hydraulic Works Department like annual precipitation or do you yourself compute them from the annual precipitation?

AC 7: Seasonal precipitation amounts were computed based on these annual precipitation amounts provided by the Environment and Hydraulic Works Department for each catchment and the seasonal distribution of precipitation in the hydro-meteorological stations listed in Table 1.

In order to clarify this in the manuscript, the original text: "Seasonal precipitation amounts for autumn (AP, mm), winter (WP, mm), spring (SpP, mm) and summer (SuP, mm) were also computed."

Was modified as follows: P5; L31: "Seasonal precipitation amounts for autumn (AP, mm), winter (WP, mm), spring (SpP, mm) and summer (SuP, mm) were computed based on the annual precipitation amounts for each catchment and the seasonal distribution of precipitation in the hydro-meteorological station listed in Table 1 for each catchment".

RC 8: P6, line 21-34: this section would be a bit more clear if you switch the paragraph starting in line 30 with that starting in line 25, since the "real values of explanatory variables" in line 23 are the land cover combinations explained starting from line 30 if I understand correctly?

AC 8: In this case, variables are precipitation and land cover types, and the next two paragraphs are explaining how those variables (precipitation first and land cover later) are considered. As in the first equation only precipitation is considered and in the second the equation is extended to land cover, the order selected was the one in the manuscript.

However, in order to clarify this, in the original text: "This allowed for a simple and direct interpretation of the influence of all variables."

The following was added: P6; L26: "This allowed for a simple and direct interpretation of the influence of all variables (precipitation and land cover types)".

RC 9: P6 line 25-29: How sensitive are your results to the choice to exclude outliers and to add precipitation of the previous study? i.e. are your conclusions different if you do not exclude outliers and / or only include precipitation of the 3-month season?

AC 9: The objective of the paper was to study the influence of land cover in a natural precipitation gradient. Including extreme values, would give us the response of catchments to extreme, very particular conditions, which, each of them, should be analyzed very carefully and deeply. Moreover, the inclusion of so particular values could bias the relationship between precipitation, land cover and hydrological services.

in order to clarify this, the original text: "To avoid considering outliers, the 1st and 3rd quartiles of the precipitation data series were calculated for the selected period (annual or seasonal) and defined as the low and high precipitation conditions."

Was slightly modified as follows: P6; L30: "To avoid biased results affected by considering extreme values, the 1st and 3rd quartiles of the precipitation data series were calculated for the selected period (annual or seasonal) and defined as the low and high precipitation conditions."

The authors do not know what the referee refers to when talking about precipitation of the previous study.

Including precipitation of only one season in the analysis does not give significant statistical results. in order to clarify this, in the original text: "..., while for seasonal scale, precipitation of the season studied plus that of the previous season (6 months total) were considered."

The following was added: P6; L32: "..., while for seasonal scale, precipitation of the season studied plus that of the previous season (6 months total) were considered. The statistical analysis was also carried out considering precipitation of the studied season (3 months), however, no statistically significant results were found."

RC 10: P6 line 30 – P7 line 5: are your 6 land cover combinations the most common ones in the region? If so how much of the area do they represent?

AC 10: The 6 land cover combinations created are based on real data, so that even if the combination itself may not exactly exist in any of the catchments as is, catchments with high percentage of exotic plantations (76 % in Aixola, Table 1); of native forests (66% in Añarbe, Table1); of paturelands (42 % in Urkulu, Table 1) exist. Table 1 shows the real combinations existing in the studied catchments and Table 2 shows the statistics for those land cover types. Additionally, in the study area section some general data for land cover in Gipuzkoa province are also included.

In this sense, the original text in the study area section: "... Forests are the main land use (73 % in 2011) (MAGRAMA, 2013). The original broad-leaved forests (oak–Quercus robur, and beech–Fagus sylvatica), presently reduced to 15 % of their original area, share space with tree plantations of rapid-growth exotic species such as Pinus radiata. These exotic species were introduced in the second half of the twentieth century as a result of government support for afforestation policies."

Was slightly modified as follows: P3; L30: "... Forests are the main land use (63 % in 2011) (MAGRAMA, 2013). The original broad-leaved forests (oak–Quercus robur, and beech–Fagus sylvatica), presently reduced to 15 % of their original area, account for 28 % of the province and share space with tree plantations of rapid-growth exotic species such as Pinus radiata. These exotic species were introduced in the second half of the twentieth century as a result of government support for afforestation policies."

*Note that a correction was also done in the forest type percentage due to incorrect data included in the previous version of the manuscript.

Results and discussion: RC 11: P8 line 20: Table 6 is mentioned before Table 5. Furthermore, you could mention in the table captions that you do not show insignificant results (now I wondered for instance why Table 6 did not include Sp90m – that results for Sp90m are insignificant is only mentioned in later sections).

AC 11: In the new version of the manuscript, and following recomentations of reviewer 1, tables 4, 5 and 6 were merged in one unique table.

About insignificant results not included, a footnote in tables 4, 5 and 6 was already included to mention not significant results are not included in those tables.

RC 12: P8 line 21: I think you refer to figure 2b (Sp50m) instead if 3b (Sp10m)

AC 12: We are very sorry for the mistake, the referee is wright, so that we changed the reference to the figure in the text

Where it said: "Regression results shown in Fig. 3b indicate that native forests,

pasturelands,..."

Now it says: P8; L33: "Regression results shown in Fig. 2b indicate that native forests, pasturelands,..."

RC 13: P9 line 14-16: what potentials are usually claimed?

AC 13: A potential of reducing high flows has for a long time been attributed to forests in the literature, and some governments have somehow applied this findings. In fact, as literally mentioned in the referenced paper (Robinson et al., 2003) "In February 1995, the Environment Ministers of France, Germany, Luxembourg and the Netherlands adopted the "Declaration of Arles" to take measures to reduce future flood risks, which include land management and forestry (WMO, 1995)".

Please also note the supplement to this comment:
https://www.hydrol-earth-syst-sci-discuss.net/hess-2018-366/hess-2018-366-AC2-supplement.pdf

**Supplement:**

[revised manuscript text omitted]

---

## Author Response (AR2)

Editor Comment:
Editor Decision: Publish subject to technical corrections (01 Oct 2018) by Nadav Peleg
Comments to the Author:
Dear Ane Zabaleta and co-authors,

I read the revised manuscript and your rebuttal letter and I am pleased to inform you that your manuscript has now been accepted for publication in HESS, subject to technical corrections (please see below).

EC 1: General: All the material that is now under 'Appendix' should appear as 'Supplementary Material' (i.e. should be separate from the main text and uploaded a separate file).

AC 1: The requested change was made and now there are not Appendices but there is a Supplementary material.

EC 2: P3, Line 20: remove the comma after slopes

AC 2: comma was removed

EC 3: P6, Line 12: catchment should be catchments

AC 3: catchment was corrected to catchments

EC 4: P10, Line 6: remove 'that'

AC 4: that was removed

EC 5: P11, Line 5: "Land cover effects on hydrological services - implications"

AC 5: the requested change was made

EC 6: P11, Line 36: add "of" river discharge…

AC 6: of was added

EC 7: P11, Line 37: add "to" achieve a solid…

AC 7: to was added

EC 8: P11, Line 37/38: remove one of the "and" in the sentence

AC 8: the first and was removed

EC 9: P11, Line 18: use "policies" instead of "politics"

AC 9: politics was changed by poticies

EC 10: P11, Line 19: Please revise the sentence and consider removing "is necessary"

AC 10: the sentence was modified as follows: "As Ellison et al. (2017) stated, the impact of land management policies on hydrological services is not usually considered. However, it is crucial to take into account local findings on the relationship between land cover and water-related ecosystem services in order to design an adequate integrated catchment management."

EC 11: In the footnote: Please add "respectively" when applicable. E.g."…Y10m, A10m, W10m, Sp10m and Su10m: 10th percentile for annual, autumn, winter, spring and summer discharge series, respectively;

AC 11: respectively was added where necessary

EC 12: P9, Line 13: "Consequently, the land cover combination that gives the highest or lowest median discharge values changes with annual and seasonal precipitation amounts ". Which land cover? In the previous sentence you referred to two combinations Nat and NAT+PAST? Which one is the land cover the sentence refers to?

AC 12: the sentence was modified to improve comprehension: "Consequently, to establish the optimal land cover combination for median discharge it should be taken into account annual and seasonal precipitation amounts."